# CONV-BASIS: A NEW PARADIGM FOR EFFICIENT ATTENTION INFERENCE AND GRADIENT COMPUTATION IN TRANSFORMERS

## ABSTRACT

The self-attention mechanism is the key to the success of transformers in recent Large Language Models (LLMs). However, the quadratic computational cost $O(n^2)$ in the input sequence length $n$ is a notorious obstacle for further improvement and scalability in longer contexts. In this work, we leverage the convolution-like structure of attention matrices to develop an efficient approximation method for attention computation using convolution matrices. We propose a conv basis system, analogous to the rank basis, and show that any lower triangular matrix can always be decomposed as a sum of structured convolution matrices in this basis. We then design a fast algorithm to approximate the attention matrix via a sum of such $k$ convolution matrices. This allows us to compute the attention *inference* via Fast Fourier Transforms (FFT) in $O(knd \log n)$ time, where $d$ is the hidden dimension, and thus achieve almost linear time $n^{1+o(1)}$ in the practical scenario where $kd = n^{o(1)}$. Furthermore, the attention *training forward* and *backward gradient* can be computed in $n^{1+o(1)}$ as well. We provide theoretical guarantees on the run time and approximation error and conduct preliminary experiments to evaluate its effectiveness. We hope our new paradigm for accelerating attention computation in transformer models can help their application to longer contexts.

## 1 INTRODUCTION

Numerous notable large language models (LLMs) in natural language processing (NLP) have emerged in these two years, such as Mistral (Jiang et al., 2023), Gemini (Team et al., 2023), Claude3 (Anthropic, 2024), GPT-4 (Achiam et al., 2023), Llama3 (AI, 2024) and so on. These models have profoundly changed the world and have been widely used in human activities, such as education (Kasneci et al., 2023), law (Sun, 2023), finance (Li et al., 2023a), bio-informatics (Thirunavukarasu et al., 2023), coding (Hou et al., 2024), and even creative writing (Achiam et al., 2023) such as top AI conference reviews (Liang et al., 2024a). The key component of the generative LLMs success is the decoder-only transformer architecture introduced by Vaswani et al. (2017). The transformer uses the self-attention mechanism, allowing the model to capture long-range dependencies in the input sequence. Self-attention computes a weighted sum of the input tokens, where the weights are determined by the similarity between each pair of tokens. This enables the model to attend to relevant information from different parts of the sequence when generating the output. However, the computational complexity of the self-attention in transformers grows quadratically $O(n^2)$ with the input length $n$, limiting their applicability to long context, e.g., 128k, 200k, 1000k input tokens for GPT4 (Achiam et al., 2023), Claude3 (Anthropic, 2024), Gemma (Team et al., 2024) respectively.

The complexity $O(n^2)$ comes from computing the similarity between each pair of tokens, which will introduce an $n \times n$ size matrix. More specifically, let $d$ be the hidden dimension and let $Q, K \in \mathbb{R}^{n \times d}$ be the query and key matrices of input. Then attention needs to compute Softmax on $QK^\top \in \mathbb{R}^{n \times n}$. Although $QK^\top$ is at most rank-$d$, $\mathsf{Softmax}(QK^\top) \in \mathbb{R}^{n \times n}$ may be full rank in Softmax attention.

To overcome the computational obstacle of $\mathsf{Softmax}(QK^\top)$, many studies propose more efficient attention computation methods that can scale gracely with the sequence length while maintaining the model's performance. Alman & Song (2023) show that if all entry of $QK^\top$ is bounded and $d = O(\log n)$, $\mathsf{Softmax}(QK^\top)$ will be "close" to a low-rank matrix. Then, they present an algorithm

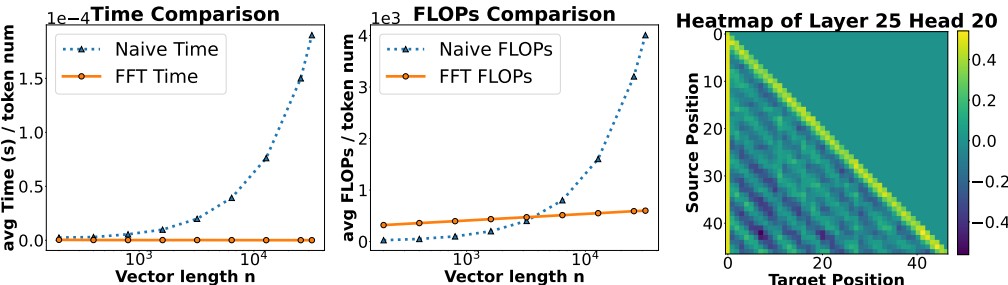

Figure 1: (a) In the left two figures, we compare the complexity of $\mathsf{conv}(a) \cdot w$ between the Naive way and FFT way, where random vector $a, w \in \mathbb{R}^n$ and $\mathsf{conv}(a) \in \mathbb{R}^{n \times n}$ (Definition 2.5). The $x$-axis is the input token number $n$. The $y$-axis is the average CPU time/Float Operations (FLOPs) over $n$, in the first/second figure. The number reported is an average of 100 runs with Numpy implementation. It is clear to see the Naive way takes $O(n^2)$ while the FFT way takes $O(n \log n)$. (b) In the right figure, we plot one $QK^\top \in \mathbb{R}^{n \times n}$ in Llama3 (AI, 2024), where input is from the SST-2 (Wang et al., 2018) with $n = 47$ tokens. It is clear to see the conv-like structure in the attention matrix.

that can approximate attention computation in almost linear time. Similarly, by uniform Softmax column norms assumption and sparse assumption, Han et al. (2024) solve attention computation in almost linear time, where they identify large entries in the attention matrix and only focus on them.

Another line of work (Olsson et al., 2022; Song & Zhong, 2023; Nichani et al., 2024; Reddy, 2024) find that the attention pattern has convolutional-like (or "diagonalized") structure (see Figure 1 (b)), mathematically, $A_{i,j} \approx A_{i',j'}$ when $i - j = i' - j'$, where we can see $i - j$ as the position distance between two tokens. It is relevant to the bag-of-words or $n$-gram concept, i.e., $n$ adjacent symbols or words in NLP. Furthermore, the convolutional-like structure can be connected to convolution recurrent models (Bai et al., 2018), Hyena Hierarchy models (Poli et al., 2023; Massaroli et al., 2023), and structured state space models (SSMs) such as Mamba (Gu & Dao, 2023). More specifically, we can use multiple convolution matrices to approximate an attention matrix, whose intuition is similar to the low-rank approximation in the sense of computation acceleration. Note that the matrix product of a convolution matrix and a vector can be computed by Fast Fourier Transform (FFT) with time complexity $O(n \log(n))$, while the naive way takes $O(n^2)$ time (see details in Figure 1 (a)). Therefore, it is natural to ask:

*Can we exploit the convolutional structure to accelerate the attention computation?*

In this paper, we use multiple convolution matrices to approximately solve the attention computation efficiently. Informally speaking, we have the following results, which can apply to **any** $Q, K \in \mathbb{R}^{n \times d}$.

**Theorem 1.1** (Main result, informal version of Theorem 3.4). *Let $\epsilon > 0$, $k \in [n]$ and $Q, K \in \mathbb{R}^{n \times d}$. If $QK^\top$ is $\epsilon$-close in $\ell_\infty$ norm to a matrix with $k$-conv basis (Definition 3.1), then we can solve the Exact Attention Computation (Definition 2.3) in $O(knd \log(n))$ time via FFT with error up to $O(\epsilon)$.*

When $kd = n^{o(1)}$, our method gets almost linear time $n^{1+o(1)}$. Similarly to the low-rank approximation, in our work, we build up a conv basis system, analogous to the rank basis, and show that any lower triangular matrix $H \in \mathbb{R}^{n \times n}$ can always be decomposed into $k$-conv basis for some $k \in [n]$, where $[n] = \{1, 2, \dots, n\}$ (Lemma 2.12 and Theorem 3.3). Then, our Algorithm 2 can quickly decompose $QK^\top$ into $k$ convolution matrix when $QK^\top$ satisfying some non-degenerate properties (see properties in Definition 3.1). Finally, via FFT, we only need time complexity $O(knd \log(n))$ to solve the task (Algorithm 1 and Theorem 3.4), while the naive methods require $O(n^2 d)$.

Thus, our algorithm can achieve attention inference in $O(knd \log(n))$, without any parameter updates, e.g., re-train or finetune. Our theorems can also applied to accelerate attention training, taking $O(knd \log n + nd^2)$ time for forward computation and $O(knd^2 \log n)$ time for backward gradient computation (Theorem 4.6). Furthermore, we conduct preliminary experiments to evaluate its effectiveness (Section 6). Additionally, our technique can also be applied to extend the low-rank

approximation of attention matrices (Alman & Song, 2023) to more general settings (Theorem 5.5). In detail, Alman & Song (2023) only works on attention approximation without an attention mask, while ours can be applied to different kinds of attention masks, including the most popular causal attention mask (Definition 2.2). This shows the broad applicability of our analysis.

**Our contributions are summarized as follows.**

- We propose a conv basis system, and show that any lower triangular matrix $H \in \mathbb{R}^{n \times n}$ can always be decomposed into $k$-conv basis for some $k \in [n]$ (Lemma 2.12 and Theorem 3.3).

- We propose an algorithm (Algorithm 2) that can quickly decompose any lower triangular matrix into its $k$ convolution basis. So via FFT, we can solve Exact Attention Computation task in $O(knd \log(n))$ (Algorithm 1 and Corollary 3.5). When $kd = n^{o(1)}$, our method takes almost linear time $n^{1+o(1)}$. Our results are beyond or comparable to previous works (see comparison below).

- During attention inference, our algorithm takes $O(knd \log(n))$, without any parameter updates, e.g., re-train or fine-tune (Theorem 3.4). Due to convolution property and Fourier analysis, our new method has a better theoretical guarantee than existing approaches.

- During attention training, our methods take $O(knd \log n + nd^2)$ time for forward computation and $O(knd^2 \log n)$ time for backward gradient computation (Theorem 4.6).

- Our broadly applicable technique can be applied to the low-rank approximation of attention matrices and extend existing results to more general settings (Theorem 5.5).

**Detailed comparison with previous works.** Our results are beyond or comparable to the two brilliant previous works. (1) To guarantee a small approximation error, for the attention matrix, Alman & Song (2023) needs bounded entries assumption and $d = O(\log n)$ assumption, while Han et al. (2024) needs uniform Softmax column norms assumption and sparse assumption. However, without all these assumptions, our algorithm can still guarantee a small approximation error (Corollary 3.5), i.e., our algorithm can apply to any $Q, K$ including unbounded matrices, dense matrices, and any hidden dimension $d$. (2) To guarantee a truly subquadratic running time, Alman & Song (2023) needs to assume $d = O(\log n)$ to get $n^{1+o(1)}$ time complexity. However, for our algorithm, as long as $d = n^{o(1)}$ and $k = n^{o(1)}$, we achieve running time $n^{1+o(1)}$. This has much less restriction on $d$. Moreover, our time complexity covers from $n^{1+o(1)}$ to $n^{2-\Omega(1)}$ with different $d$, while Alman & Song (2023) can only handle $d = O(\log n)$. (3) To guarantee a truly subquadratic running time, Han et al. (2024) needs to assume $dm = n^{2-\Omega(1)}$, as they get $O(dn^{1+o(1)} + dm)$ time complexity where $m$ is the number of large entries in attention matrices. Our work gets $O(knd \log(n))$ time complexity and we need $kd = n^{1-\Omega(1)}$ to get truly subquadratic running time. For the situation $m = n^{1+o(1)}, d = n^{o(1)}$ and $k = n^{o(1)}$, both our algorithm and Han et al. (2024) run in $n^{1+o(1)}$ time. For the situation $m = n^{1+\Omega(1)}, d = n^{o(1)}$ and $k = n^{o(1)}$, running time in Han et al. (2024) will be truly super-linear $n^{1+\Omega(1)}$ while our algorithm remains almost $n^{1+o(1)}$ linear time[1].

## 1.1 RELATED WORK

**Attention matrix conv-like structure.** Very recent works study the conv-like attention matrix. Elhage et al. (2021); Olsson et al. (2022) find that in-context learning is driven by the formation of "induction heads"–attention heads that copy patterns from earlier in the input sequence. This is reflected in the attention matrix becoming more diagonal, with tokens attending primarily to preceding tokens that match the current token. In Song & Zhong (2023) Figure 6, they show a similar conv-like attention pattern for other important attention circuits. Figure 3 of Reddy (2024) shows that in a minimal classification task, the abrupt emergence of in-context learning coincides with the formation of an induction head, characterized by a diagonal attention pattern. Nichani et al. (2024) proves that for a simplified task, gradient descent causes a transformer to encode the causal graph structure of the task in the attention matrix. This results in tokens attending primarily to their causal parents reflected in a sparse diagonal structure (Figure 2). In Li et al. (2024a), the conv-like attention matrix can also be observed when learning math tasks. Moreover, Cai et al. (2024) uses convolutional kernels to compress the KV-cache size for fast LLM generation.

---

[1]Considering the case where attention matrix is all 1 lower triangular matrix, we have $k = 1$ and $m = n(n+1)/2$.

## 2 PRELIMINARY

In Section 2.1, we introduce the basic definitions and mathematical properties. In Section 2.2, we give the formal definition of the sub-convolution matrix and present it basic properties.

**Notations.** We use $\circ$ to denote element-wise multiplication. We denote $[n] = \{1, 2, \ldots, n\}$ and $[0]$ as an empty set. We denote $\mathbf{0}_n$ and $\mathbf{1}_n$ as the $n$-dimensional vector whose entries are all $0$ and $1$ respectively. We denote $\exp(\cdot)$ as the element-wise exponential function. We denote $[x_a, x_{a+1}, \ldots, x_b]^\top \in \mathbb{R}^{b-a+1}$ as $x_{a:b}$, where $1 \leq a \leq b \leq n$, similarly for matrix. Let $\mathrm{diag} : \mathbb{R}^n \to \mathbb{R}^{n \times n}$ be defined as $\mathrm{diag}(x)_{i,i} = x_i$ and $\mathrm{diag}(x)_{i,j} = 0$, for all $i \neq j$. For a matrix $A \in \mathbb{R}^{m \times n}$, we define its $\ell_1$ norm as $\|A\|_1 = \sum_{i=1}^{m} \sum_{j=1}^{n} |A_{ij}|$, $\ell_\infty$ norm as $\|A\|_\infty = \max_{i,j} |A_{ij}|$, and Frobenius norm as $\|A\|_F := \sqrt{\sum_{i,j} A_{i,j}^2}$, where $A_{ij}$ is an entry at the $i$-th row and $j$-th column.

### 2.1 BASIC DEFINITIONS AND FACTS ABOUT ATTENTION AND conv

Now, we present basic definitions. We start by introducing the input and weight matrix.

**Definition 2.1** (Input and weight matrix). *We define the input sequence as $X \in \mathbb{R}^{n \times d}$ and the key, query, and value weight matrix as $W_K, W_Q, W_V \in \mathbb{R}^{d \times d}$. Then, we define the key, query, and value matrix as $K := XW_K \in \mathbb{R}^{n \times d}$, $Q := XW_Q \in \mathbb{R}^{n \times d}$, $V := XW_V \in \mathbb{R}^{n \times d}$.*

It is straightforward to see $QK^\top = XW_Q W_K^\top X^\top$. In generative LLMs, there is a causal attention mask $M$ to guarantee the later tokens cannot see the previous tokens during generation.

**Definition 2.2** (Causal attention mask). *We define the causal attention mask as $M \in \{0, 1\}^{n \times n}$, where $M_{i,j} = 1$ if $i \geq j$ and $M_{i,j} = 0$ otherwise. We define $M_j$ be the $j$-th column of $M$.*

Now, we introduce the mathematical definition of the exact attention computation with a mask.

**Definition 2.3** (Exact attention computation). *Let $Q, K, V \in \mathbb{R}^{n \times d}$ be the query, key, and value matrices respectively defined in Definition 2.1. Let $M \in \{0, 1\}^{n \times n}$ be the attention mask defined in Definition 2.2. The goal of the Exact Attention Computation is to find the matrix $\mathrm{Att}(M, Q, K, V) \in \mathbb{R}^{n \times d}$, which is defined as*

$$\mathrm{Att}(M, Q, K, V) := D^{-1}AV$$

*where $A \in \mathbb{R}^{n \times n}$ is a lower triangular matrix and $D \in \mathbb{R}^{n \times n}$ is a diagonal matrix, i.e., $A := M \circ \exp(QK^\top)$ and $D := \mathrm{diag}(A\mathbf{1}_n)$.*

**Remark 2.4.** *In Definition 2.3, we divide the Softmax operation into an element-wise exp operation and a diagonal normalization matrix $D$ to obtain a clear formulation.*

Efficiently computing the attention needs to exploit structured matrices that enable fast multiplication algorithms. Here, we define the convolution matrix, which is a structured matrix where each row vector is rotated one element to the right relative to the preceding row vector.

**Definition 2.5** (Convolution matrix). *Let $a \in \mathbb{R}^n$. We define $\mathsf{conv} : \mathbb{R}^n \to \mathbb{R}^{n \times n}$ as,*

$$\mathsf{conv}(a) := \begin{bmatrix} a_1 & 0 & 0 & \cdots & 0 \\ a_2 & a_1 & 0 & \cdots & 0 \\ a_3 & a_2 & a_1 & \cdots & 0 \\ \vdots & \vdots & \vdots & \ddots & \vdots \\ a_n & a_{n-1} & a_{n-2} & \cdots & a_1 \end{bmatrix}.$$

By the following fact, we know that the rank of a convolution matrix can be an arbitrary number. Thus, our conv-basis is totally different from the rank basis. See proof in Appendix B.1.

**Claim 2.6.** *We have $\mathsf{conv}(e_j) \in \mathbb{R}^{n \times n}$ is a $j$-rank matrix, where the $j$-th entry of $e_j \in \mathbb{R}^n$ is $1$ and all other entries are $0$.*

Efficient computation of the convolution operation is crucial for many applications. The convolution theorem states that the circular convolution of two vectors can be computed efficiently using the Fast Fourier Transform (FFT). This leads to the following claim (see proof in Appendix B.1):

**Claim 2.7.** *Let* conv *be defined in Definition 2.5. For any* $a, x \in \mathbb{R}^n$*,* conv$(a)x$ *can be computed in* $O(n \log n)$ *via FFT.*

One property of convolution matrices is that they are additive with respect to the input vectors. In other words, the convolution of the sum of two vectors is equal to the sum of the convolutions of the individual vectors. This is stated formally in the following claim (see proof in Appendix B.1):

**Claim 2.8.** conv *is additive, i.e., for any* $a, b, x \in \mathbb{R}^n$ *we have* conv$(a)x$+conv$(b)x$ = conv$(a+b)x$.

Many other interesting facts and properties about the convolution matrix are used in our main theorem proof. Due to space limitations, we leave them in Appendix B.1 for reader interests.

## 2.2 SUB-CONVOLUTION MATRIX: DEFINITIONS AND PROPERTIES

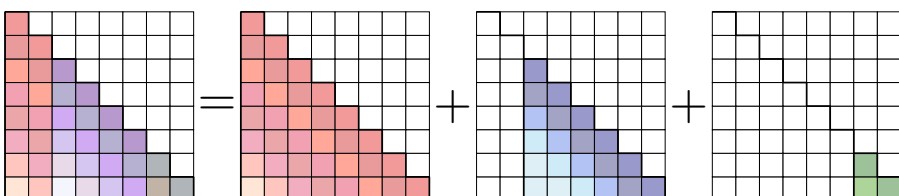

Figure 2: A matrix with 3-conv basis. We present an example of the matrix defined in Definition 2.11 when $k = 3$. The matrix with 3-conv basis is on the left-hand side of the equation in this figure. The red entries in this matrix come from the first matrix on the right-hand side. The purple entries in this matrix are the sum of the red entries from the first matrix on the right-hand side and the blue entries from the second matrix on the right-hand side. The dark green entries are equal to the sum of red, green, and blue entries from the matrices on the right-hand side.

If we would like to use conv as a basis system, we need to introduce some new concepts. Recall that, in general, the sum of two rank-1 matrices is a rank-2 two matrix. Due to conv being additive, the sum of two convolution matrices is another convolution matrix, which does not hold the above property. Thus, we need to introduce sub-convolution matrices to be the basis.

**Definition 2.9** (Sub-convolution matrix). *Let* $m \in [n]$*. For any* $a \in \mathbb{R}^n$*. We define the sub-convolution matrix* conv$(a, m)$ *as*

$$\mathsf{conv}(a, m) = \begin{bmatrix} \mathbf{0}_{(n-m)\times(n-m)} & \mathbf{0}_{(n-m)\times m} \\ \mathbf{0}_{m\times(n-m)} & \mathsf{conv}(a_{1:m}) \end{bmatrix}.$$

*Given two vectors* $a, x \in \mathbb{R}^n$*, let* $a *_m x \in \mathbb{R}^n$ *denote the sub-convolution operator between* $a$ *and* $x$*, i.e.,* conv$(a, m)x = a *_m x$.

Similarly, sub-convolution can be computed in $O(n \log n)$ time via FFT (see proof in Appendix B.1).

**Claim 2.10.** *Let* $m \in [n]$*. For any* $a, x \in \mathbb{R}^n$*,* conv$(a, m)x$*, (defined in Definition 2.9) can be computed in* $O(n \log n)$ *via FFT.*

Here, we present the definition of the matrix with $k$-conv basis which is non-reducible.

**Definition 2.11** (Matrix with $k$-conv basis). *Let* $k \in [n]$*. We say a lower triangular matrix* $H \neq \mathbf{0}_{n \times n} \in \mathbb{R}^{n \times n}$ *has* $k$-conv *basis if*

- *There exists* $b_1, \ldots, b_k \in \mathbb{R}^n$ *and* $k$ *integers* $m_1, m_2, \ldots, m_k$ *satisfying* $n \geq m_1 > m_2 > \cdots > m_k \geq 1$ *such that* $H = \sum_{i \in [k]} \mathsf{conv}(b_i, m_i)$*, (defined in Definition 2.9).*

- *For any* $b_1, \ldots, b_{k-1} \in \mathbb{R}^n$ *and* $k - 1$ *integers* $m_1, m_2, \ldots, m_{k-1}$ *satisfying* $n \geq m_1 > m_2 > \cdots > m_{k-1} \geq 1$ *we have* $H \neq \sum_{i \in [k-1]} \mathsf{conv}(b_i, m_i)$*.*

The following lemma establishes that any non-zero lower triangular matrix can be represented as a matrix with a $k$-conv basis for some unique $k$ between 1 and $n$. The proof is in Appendix E.1.

**Lemma 2.12.** *For any lower triangular matrix* $H \neq \mathbf{0}_{n \times n} \in \mathbb{R}^{n \times n}$*, there exists a unique* $k \in [n]$ *such that* $H$ *is a matrix with* $k$-conv *basis.*

# 3 conv APPROXIMATION DURING INFERENCE

In Section 3.1, we introduce the basic definitions to support our algorithmic analysis in this section. In Section 3.2, we present the binary search and recover $k$-conv algorithms and present their theoretical guarantees. In Section 3.3, we provide the formal version of our main result.

## 3.1 KEY CONCEPTS

Any non-zero lower triangular matrix can be represented as a matrix with a $k$-conv basis for some unique $k$ between 1 and $n$ (Lemma 2.12). However, exactly getting $k$ is hard and the definition is too strict for the algorithm design. Thus, for more flexibility, we introduce a more general definition of non-degenerate $k$-conv basis as below, which is a proxy notion to relax the conditions required.

**Definition 3.1** (Non-degenerate $k$-conv basis). *Let $T \in [n]$, $\delta \geq 0$, and $k \in [n + 1 - T]$. Let $b_1, \ldots, b_k \in \mathbb{R}^n$ and $k$ integers $m_1, m_2, \ldots, m_k$ satisfying $n \geq m_1 > m_2 > \cdots > m_k \geq T$. Let $H = \sum_{i \in [k]} \mathsf{conv}(b_i, m_i)$. If for each basis $i \in [k]$, for all $j \in [i]$, we have $\|\sum_{l=j}^{i}(b_l)_{1:T}\|_1 \geq \delta$, then we define $H \in \mathbb{R}^{n \times n}$ to be a matrix with $(T, \delta)$-non-degenerate $k$-conv basis.*

Here $(T, \delta)$-non-degenerate $k$-conv basis means that each conv basis cannot be "covered" by the other basis easily.

**Definition 3.2.** *We define $G$ as a $\epsilon$-close $(T, \delta)$-non-degenerate $k$-conv basis matrix when $G = H + R$, where $H$ is a $(T, \delta)$-non-degenerate $k$-conv basis matrix defined in Definition 3.1 and the noise matrix $R \in \mathbb{R}^{n \times n}$ satisfies $\|R\|_\infty \leq \epsilon \leq \frac{\delta}{5T}$.*

The following theorem establishes that any non-zero lower triangular matrix can be represented as an $\epsilon$-close $(T, \delta)$-non-degenerate $k$-conv basis matrix (see proof in Section B.2). There may be many different choices of $(k, T, \delta, \epsilon)$, which provide flexibility for our Algorithm 1.

**Theorem 3.3.** *For any lower triangular matrix $G \neq \mathbf{0}_{n \times n} \in \mathbb{R}^{n \times n}$, there exists $k, T \in [n]$ and $\delta, \epsilon \geq 0$ such that $G$ is a $\epsilon$-close $(T, \delta)$-non-degenerate $k$-conv basis matrix.*

## 3.2 ALGORITHMS AND THEIR PROPERTIES

Now, we present our main Algorithm 1. We present Algorithm 2 and Algorithm 3 as well.

---

**Algorithm 1** Main $k$-conv forward

1: **procedure** convFORWARD($Q, K, V \in \mathbb{R}^{n \times d}, k, T \in [n], \delta, \epsilon \in \mathbb{R}_{\geq 0}$)     ▷ Theorem 3.4
2:     $\widetilde{b}_1, \ldots, \widetilde{b}_k, m_1, \ldots, m_k \leftarrow \mathrm{RECOVER}(Q, K, k, T, \delta, \epsilon)$    ▷ Algorithm 2, recover $k$-conv
3:     $\widetilde{D} \leftarrow \mathrm{diag}(\sum_{r \in [k]} \mathsf{conv}(\widetilde{b}_r, m_r)\mathbf{1}_n)$ by FFT in Claim 2.10
4:     $\widetilde{Y} \leftarrow \widetilde{D}^{-1} \sum_{r \in [k]} \mathsf{conv}(\widetilde{b}_r, m_r)V$ by FFT in Claim 2.10
5:     **return** $\widetilde{Y}$
6: **end procedure**

---

In Algorithm 1, we first using Algorithm 2 to get $k$ conv basis. Then, we can get the approximated normalization matrix $\widetilde{D}$ and the final output $\widetilde{Y}$ by FFT in Claim 2.7.

In Algorithm 2, we iteratively use binary search (Algorithm 3) to find the conv basis position and calculate their values. Note that, in the end, we need to change $b'_i$ to $\widetilde{b}_i$ by incorporating $\exp$ function used in the Softmax. We will provide proof of correctness and complexity in the following section.

In Algorithm 3, we use binary search to efficiently locate the convolution basis position by leveraging the non-degenerate property (see Definitions 3.1 and 3.2) of the attention matrix. This allows us to find $k$-conv-basis in our main Algorithm 1, enabling better control over the running time while bounding the error. The choice of $k$ thus balances the trade-off between accuracy and efficiency. Technically, Algorithm 3 identifies positions in the attention matrix where the $\ell_1$ norm of remaining attention values exceeds the threshold $\delta - 2T\epsilon$. The non-degenerate property enables the binary search algorithm to find the next convolution basis position in $O(\log n)$ steps.

---

**Algorithm 2** Recover $k$-conv

1: **procedure** RECOVER($Q, K \in \mathbb{R}^{n \times d}, k, T \in [n], \delta, \epsilon \in \mathbb{R}_{\geq 0}$)
2:      $v \leftarrow \mathbf{0}_T, u \leftarrow \mathbf{0}_n, s \leftarrow 0, t \leftarrow n - T + 1$          ▷ Initialize the state for binary search
3:      **for** $i = 1 \rightarrow k$ **do**
4:          $s \leftarrow s + 1$
5:          $s \leftarrow$ SEARCH($Q, K, k, T, \delta, \epsilon, v, s, t$) ▷ Algorithm 3 in Appendix B.2, binary search the next conv basis position
6:          $m_i \leftarrow n - s + 1$
7:          $\widetilde{H}_s \leftarrow M_s \circ (Q(K^\top)_s)$
8:          $(b'_i)_{1:m_i} \leftarrow \widetilde{H}_{s,s:s+m_i-1} - u_{1:m_i}, (b'_i)_{m_i+1:n} \leftarrow \mathbf{0}_{n-m_i}$      ▷ Get the conv basis value
9:          $v \leftarrow v + (b'_i)_{1:T}$
10:         $u \leftarrow u + b'_i$
11:      **end for**
12:      Get $\widetilde{b}_1, \ldots, \widetilde{b}_k$ by Lemma B.16 from $b'_1, \ldots, b'_k$ and $m_1, \ldots, m_k$
13:      **return** $\widetilde{b}_1, \ldots, \widetilde{b}_k, m_1, \ldots, m_k$
14: **end procedure**

---

**Algorithm 3** Binary search

1: **procedure** SEARCH($Q, K \in \mathbb{R}^{n \times d}, k, T \in [n], \delta, \epsilon \in \mathbb{R}_{\geq 0}, v \in \mathbb{R}^T, s, t \in [n]$)
2:      **if** $s \geq t$ **then**
3:          **return** $s$
4:      **end if**
5:      $j \leftarrow \lfloor (s + t)/2 \rfloor$
6:      $\widetilde{H}_j \leftarrow M_j \circ (Q(K^\top)_j)$          ▷ $j \in [n], M$ is attention mask defined in Definition 2.2
7:      $\alpha \leftarrow \|(\widetilde{H}_j)_{j:j+T-1} - v\|_1$
8:      **if** $\alpha \geq \delta - 2T\epsilon$ **then**
9:          **return** SEARCH($Q, K, k, T, \delta, \epsilon, v, s, j$)
10:      **else**
11:          **return** SEARCH($Q, K, k, T, \delta, \epsilon, v, j + 1, t$)
12:      **end if**
13: **end procedure**

---

### 3.3 MAIN THEORETICAL RESULT

In this section, we present our main result.

**Theorem 3.4** (Main conv results for inference). *Let $Q, K, V \in \mathbb{R}^{n \times d}$. Recall $A = M \circ \exp(QK^\top) \in \mathbb{R}^{n \times n}$, $D = \mathrm{diag}(A\mathbf{1}_n) \in \mathbb{R}^{n \times n}$ defined in Definition 2.3. We denote $Y := D^{-1}AV \in \mathbb{R}^{n \times d}$. Let $M \circ (QK^\top)$ be a $\epsilon$-close $(T, \delta)$-non-degenerate $k$-conv basis matrix as defined in Definition 3.2, where $\delta, \epsilon \geq 0$ and $k, T \in [n]$. By Algorithm 1, we can get $\widetilde{Y}$ such that*

$$\|Y - \widetilde{Y}\|_\infty \leq 2(\exp(2\epsilon) - 1)\|V\|_\infty,$$

*whose time complexity is $O(knd \log(n))$ given $M, Q, K, V$.*

*Proof sketch of Theorem 3.4.* See complete proof in Appendix B.4. The proof idea is that using binary search to recover all non-degenerate conv basis (Lemma B.19), which takes $O(knd \log(n))$ time and has upto $2(\exp(2\epsilon) - 1)\|V\|_\infty$ error (Lemma B.20). Then, via FFT (Claim 2.10), we finish the proof.          $\square$

Note that our algorithm can handle any $Q, K \in \mathbb{R}^{d \times d}$. Furthermore, we can exactly recover $Y$ if we do not care about the time complexity. We formally describe the above intuition in the following.

**Corollary 3.5** (Exact conv inference). *Let $Q, K, V \in \mathbb{R}^{n \times d}$. Recall $A = M \circ \exp(QK^\top) \in \mathbb{R}^{n \times n}$, $D = \mathrm{diag}(A\mathbf{1}_n) \in \mathbb{R}^{n \times n}$ defined in Definition 2.3. We denote $Y := D^{-1}AV \in \mathbb{R}^{n \times d}$. For any $\epsilon \geq 0$ and any $Q, K, V$, there exists hyper-parameter $k, T \in [n]$ and $\delta \geq 0$ such that Algorithm 1*

can output $\widetilde{Y}$ satisfying $\|Y - \widetilde{Y}\|_\infty \leq 2(\exp(2\epsilon) - 1)\|V\|_\infty$. *Furthermore, we can exactly get* $Y$, *i.e.,* $\epsilon = 0$, *through Algorithm 1 with time complexity* $O(n^2 d \log(n))$ *in the worst case.*

See proof of the above corollary in Appendix B.4. By Theorem 3.4, when $\epsilon = O(1)$, we directly get the attention inference time complexity is $O(knd \log(n))$ with error up to $O(\epsilon)$ as claimed in Section 1. It may enable further improvement and scalability of LLMs in the longer context.

Moreover, in Appendix A, we provide a detailed discussion about two case studies, LongLora (Chen et al., 2023b) and RoPE (Su et al., 2024), where our algorithm can apply to these two long-context LLMs as well. We also provide further discussion on limitations and extensions there.

## 4 conv APPROXIMATION FOR TRAINING

We can apply our algorithm to accelerate attention training including forward and back propagation. We first define the attention training task, which is also used in Alman & Song (2024a).

**Definition 4.1** (Attention optimization). *Given* $A_1, A_2, A_3, E \in \mathbb{R}^{n \times d}$ *and* $Y \in \mathbb{R}^{d \times d}$. *we let* $M \in \mathbb{R}^{n \times n}$ *be a casual attention mask defined in Definition 2.2. We define the optimization as*

$$\min_{X \in \mathbb{R}^{d \times d}} L(X) := 0.5\|D(X)^{-1}M \circ \exp(A_1 X A_2^\top)A_3 Y - E\|_F^2.$$

*Here* $D(X) \in \mathbb{R}^{n \times n}$ *is* $D(X) := \mathrm{diag}(M \circ \exp(A_1 X A_2^\top)\mathbf{1}_n)$.

**Remark 4.2.** *Our Attention Optimization task in Definition 4.1 covers both the cross-attention and self-attention setting. Let weight matrices* $W_K, W_Q, W_V \in \mathbb{R}^{d \times d}$ *be defined in Definition 2.1. For the self-attention setting, we can see* $A_1, A_2, A_3 \in \mathbb{R}^{n \times d}$ *as* $X \in \mathbb{R}^{n \times d}$ *in Definition 2.1, see* $X \in \mathbb{R}^{d \times d}$ *in Definition 4.1 as* $W_Q W_K^\top \in \mathbb{R}^{d \times d}$ *and see* $Y \in \mathbb{R}^{d \times d}$ *as* $W_V \in \mathbb{R}^{d \times d}$. *To overcome the quadratic complexity obstacle, we only need to handle the gradient computation of* $W_Q W_K^\top$.

Let $x, y \in \mathbb{R}^{d^2}$ denote the vectorization of $X, Y \in \mathbb{R}^{d \times d}$. Then, we define some basic notions used.

**Definition 4.3.** $\mathcal{T}_{\mathrm{mat}}(n, d, k)$ *represents the time of an* $n \times d$ *matrix times a* $d \times k$ *matrix.*

**Definition 4.4** ($\otimes$ Kronecker product). *Given two matrices* $A_1 \in \mathbb{R}^{n_1 \times d_1}$, $A_2 \in \mathbb{R}^{n_2 \times d_2}$, *we define* $A := A_1 \otimes A_2 \in \mathbb{R}^{n_1 n_2 \times d_1 d_2}$ *as follows*

$$A_{i_1+(i_2-1)n_1, j_1+(j_2-1)d_1} = (A_1)_{i_1,j_1} \cdot (A_2)_{i_2,j_2}, \quad \forall i_1 \in [n_1], i_2 \in [n_2], j_1 \in [d_1], j_2 \in [d_2].$$

Recall that during inference, we have the $n \times n$ size matrix $QK^\top$. Similarly, in gradient calculation, we have an $n \times n$ size matrix, and we denote it as $u(x)$.

**Definition 4.5.** *Let* $M \in \mathbb{R}^{n \times n}$ *be a casual attention mask defined in Definition 2.2. Let* $A_1, A_2 \in \mathbb{R}^{n \times d}$. *Suppose that* $\mathsf{A} = A_1 \otimes A_2 \in \mathbb{R}^{n^2 \times d^2}$. *For all* $j_0 \in [n]$, *let* $\mathsf{A}_{j_0} \in \mathbb{R}^{n \times d^2}$ *be the* $j_0$-th block of $\mathsf{A}$ *and* $u(x)_{j_0} := M_{j_0,*} \circ \exp(\mathsf{A}_{j_0} x)$. *Define* $u(x) \in \mathbb{R}^{n \times n}$ *as the matrix where the* $j_0$-th row corresponds to $(u(x)_{j_0})^\top$.

Then, we are ready to present our main results for attention training.

**Theorem 4.6** (Main conv result for training forward and backward gradient). *If* $u(x)$ *is a* $1/\mathrm{poly}(n)$-close $(T, \delta)$-non-degenerate $k$-conv basis matrix as defined in Definition 3.2, where $\delta \geq 0$ and $k, T \in [n]$. *Then there are algorithms that run to compute* ***training forward*** *in time* $O(knd \log n + \mathcal{T}_{\mathrm{mat}}(n, d, d))$ *and* ***backward gradient*** *in time* $O(d^2 kn \log n)$ *of attention loss (Definition 4.1) approximately up to* $1/\mathrm{poly}(n)$ *error under* $\ell_\infty$ *norm.*

*Proof sketch of Theorem 4.6.* See complete proof in Appendix C.4. During backward computation, we can convey the properties of low-rank and convolution at the same time (Lemma C.13 and Lemma C.15). Then, by tensor trick, we can compute the attention gradient based on attention inference (Lemma C.9). We finish the proof by Theorem 3.4. $\square$

**Remark 4.7.** *Note that Alman & Song (2024a) only needs to convey the low-rank property, while we need to convey the properties of low-rank and convolution simultaneously, a more general analysis.*

Our Theorem 4.6 shows that our algorithm can accelerate Transformer training as well. It may save time, resources, and energy for nowadays LLMs training.

# 5 LOW RANK APPROXIMATION

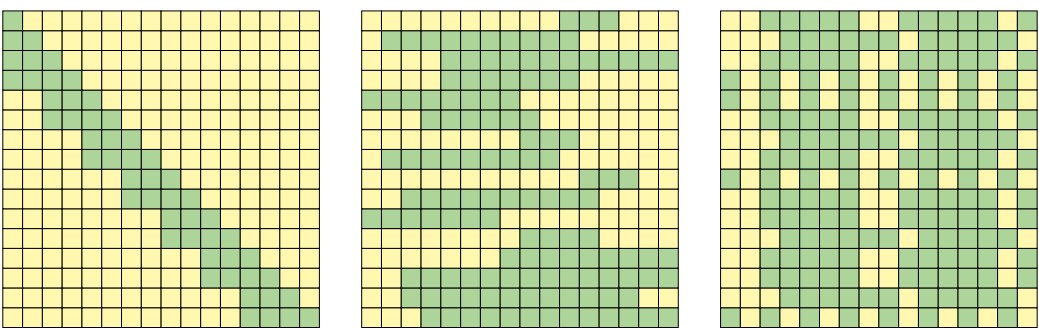

Figure 3: A $16 \times 16$ matrix with, left - row change by amortized constant mask (Definition 5.1); middle - continuous row mask (Definition 5.2); right - distinct 3 rows mask (Definition 5.4). Green means $1$ and yellow means $0$.

We can apply our analysis technique to a low-rank approximation setting in Alman & Song (2023), which only works on attention approximation without an attention mask. Equipped with our mask analysis trick, we can generalize their results with different kinds of attention masks including the most popular causal attention mask. We first introduce some practical attention masks.

**Definition 5.1.** *Let $B_j \in \mathbb{Z}_{\geq 0}$. We define the row change by amortized constant mask as $W \in \{0,1\}^{n \times n}$, where let $(W^\top)_0 = \mathbf{0}_n$ and $\|(W^\top)_j - (W^\top)_{j-1}\|_1 \leq B_j$ for any $j \in [n]$ and $(W^\top)_j$ is the $j$-th row of $W$.*

**Definition 5.2.** *We define the continuous row mask as $W \in \{0,1\}^{n \times n}$, where for each $i \in [n]$, we are given $s_i, t_i \in [n]$ such that $W_{i,j} = 1$ if $s_i \leq j \leq t_i$ and $W_{i,j} = 0$ otherwise.*

**Definition 5.3.** *We define $W \in \{0,1\}^{n \times n}$ as the distinct $r$ columns mask satisfying the following condition. Let $S_1, \cdots, S_r \subseteq [n]$ denote $r$ disjoint subsets and $\cup_{j \in [r]} S_j = [n]$. For any two $i, i' \in S_j$, we have $W_{*,i} = W_{*,i'} \in \mathbb{R}^n$, where $W_{*,i} \in \mathbb{R}^n$ denote the $i$-th column of $W \in \mathbb{R}^{n \times n}$.*

**Definition 5.4.** *We define $W \in \{0,1\}^{n \times n}$ as the distinct $r$ rows mask satisfying the following condition. Let $S_1, \cdots, S_r \subseteq [n]$ denote $r$ disjoint subsets and $\cup_{j \in [r]} S_j = [n]$. For any two $i, i' \in S_j$, we have $W_{i,*} = W_{i',*} \in \mathbb{R}^n$, where $W_{i,*} \in \mathbb{R}^n$ denotes the $i$-th row of $W \in \mathbb{R}^{n \times n}$.*

Then, we have the following main results for the low-rank setting. The proof is in Appendix D.2.

**Theorem 5.5** (Main low-rank result). *Assume the same condition as Lemma D.2. Let $\epsilon \in (0, 0.1)$. Let $Q, K, V \in \mathbb{R}^{n \times d}$. Let $U_1, U_2 \in \mathbb{R}^{n \times k}$ be defined in Lemma D.2. Let $W \in \{0,1\}^{n \times n}$ denote a mask matrix. Let $H = \exp(QK^\top/d) \in \mathbb{R}^{n \times n}$, $A = W \circ H \in \mathbb{R}^{n \times n}$ and $D = \mathrm{diag}(A\mathbf{1}_n) \in \mathbb{R}^{n \times n}$. We denote $Y := D^{-1}AV \in \mathbb{R}^{n \times d}$. Let $\widetilde{A} := W \circ U_1 U_2^\top$ and $\widetilde{D} := \mathrm{diag}(\widetilde{A}\mathbf{1}_n)$. We denote $\widetilde{Y} := \widetilde{D}^{-1}\widetilde{A}V \in \mathbb{R}^{n \times d}$. Then, we have $\|Y - \widetilde{Y}\|_\infty \leq 4\epsilon\|V\|_\infty$. The time complexity to get $\widetilde{Y}$ is*

- *$O(knd)$ when $W$ is a causal mask defined in Definition 2.2.*

- *$O(kd\sum_{j=1}^{n} B_j)$ when $W$ is a row change mask defined in Definition 5.1.*

- *$O(knd\log(n))$ when $W$ is a continuous row mask defined in Definition 5.2.*

- *$O(rnd)$ when $W$ is a distinct $r$ columns / rows mask defined in Definition 5.3 / Definition 5.4.*

Our Theorem 5.5 has the same error guarantee as Alman & Song (2023). For the normal mask, e.g., casual attention mask (Definition 2.2), Theorem 5.5 shares the same time complexity as theirs.

# 6 EXPERIMENTS

In this section, we provide our experimental results for convolution attention computing in language models, offering empirical backing to our theoretical claims.

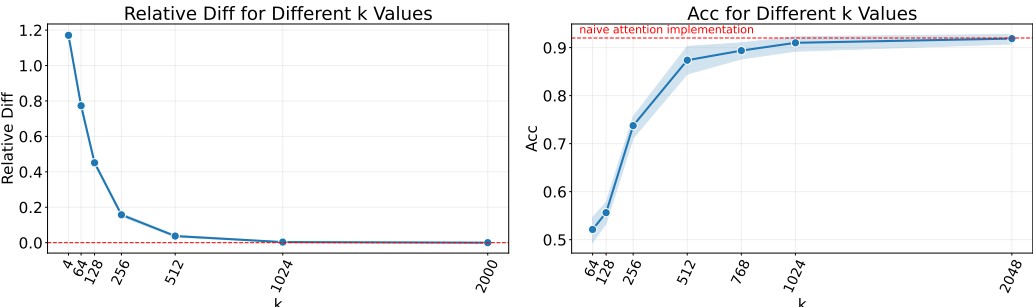

Figure 4: The comparison between the Llama3 8B Instruct with or without using our Algorithm 1 on the IMDB dataset. The input sequence length $n = 2048$. The $x$-axis is the number of conv basis. The $y$ axis is relative difference $\frac{\|Y - \widetilde{Y}\|_F^2}{\|Y\|_F^2}$ for the left figure and classification accuracy for the right figure. Note that $k = 2048$ represents the baseline of the original model, as this is the input sequence length.

**Setup.** We utilized the latest Llama3 8B Instruct model[2] (AI, 2024) as our foundation, modifying its attention mechanism with our convolution-based approach using varying numbers of convolution bases ($k$). We used the IMDB dataset (Maas et al., 2011) of labeled movie reviews. Our assessments employ two key metrics: (1) the relative difference for our final layer output $\widetilde{Y}$ and the original model's output $Y$, i.e., $\|Y - \widetilde{Y}\|_F^2 / \|Y\|_F^2$; (2) the classification accuracy. This dual approach allowed us to evaluate both the internal representations and the overall predictive performance of our convolution-based attention compared to the standard mechanism.

**Implementation details.** To ensure a fair comparison and prevent memory issues, we set the model's context length to 2048 tokens and incrementally increased the number of conv basis $k$. Note that when $k = 2048$, our convolution attention produces an identical output to the original attention mechanism. We employed an instruction-based approach to evaluate generation accuracy, formatting our input as *Review: <REVIEW> Question: Is this review positive or negative? Answer:*. This methodology allowed us to systematically assess the performance of our convolution-based attention across various complexity levels while maintaining comparability with the original model. We randomly sample 5 sample groups, with 200 samples per group, and report the results average across each group.

**Results.** The left plot in Figure 4 shows that as the base number $k$ increases, the relative MSE decreases rapidly, even with a relatively small number of bases such as $k = 256$ or $512$. This indicates that our convolution-based approach converges towards the performance of the original attention mechanism as $k$ grows. The right plot demonstrates that the accuracy of our model improves significantly as $k$ increases, and can achieve comparable accuracy to the original with $k = 512$, suggesting that our method can maintain high performance while reducing computational complexity. The results imply that our proposed method may effectively approximate the original attention mechanism, offering a promising trade-off between accuracy and efficiency, especially for scenarios where resource constraints are a concern.

## 7 CONCLUSION

We presented a novel approach for efficient attention computation in transformers using convolution matrices. Our algorithm achieves nearly linear time complexity for attention inference and gradient computation, providing better theoretical guarantees than existing methods. This work opens up a new paradigm for accelerating attention computation, enabling the application of transformers to longer contexts and potentially leading to further improvements in large language models.

---

[2]https://huggingface.co/meta-llama/Meta-Llama-3-8B-Instruct

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

# Appendix

## CONTENTS

**Roadmap.**   In Section A, we discuss two case studies: Longlora and RoPE, and provide further discussions. In Section B, we present additional details and proofs related to the convolution approximation approach. In Section C, we introduce the conv approximation in gradient. In Section D, we include supplementary material for the low-rank approximation. In Section E, we present a collection of useful tools and lemmas that are referenced throughout the main text and the appendix.

## A   FURTHER DISCUSSION

**LongLora.**   Our conv and low-rank approximation can be applied to LongLora Chen et al. (2023b), whose mask is shown in the left of Figure 3. They use this kind of sparse mask to extend the context sizes of pre-trained large language models, with limited computation cost, e.g., extending Llama2 70B from 4k context to 32k on a single $8\times$ A100 machine. As the "diagonalized" mask structure, we can directly apply our Algorithm 1 by replacing the causal attention mask (Definition 2.2) with their sparse mask for the conv approximation with time complexity $O(knd\log(n))$. Similarly, for the low-rank approximation, we directly use the second statement in Theorem 5.5 by considering row change by amortized constant mask defined in Definition 5.1 with time complexity $O(knd)$, where $B_j = O(1)$ for any $j \in [n]$.

**RoPE.**   The Rotary Position Embedding (RoPE) Su et al. (2024) designs a rotation matrix $R^{(m)} \in \mathbb{R}^{d \times d}$, for all $m \in [n]$, which can effectively encode the positional information into embedding $Q, K \in \mathbb{R}^{n \times d}$. In detail, let $q_i, k_j \in \mathbb{R}^d$, where $q_i^\top$ and $k_j^\top$ be the $i$-th and $j$-th row of $Q, K$ respectively, for any $i, j \in [n]$. By the property of rotation matrix, we have

$$(R^{(i)}q_i)^\top (R^{(j)}k_j) = q_i^\top R^{(j-i)}k_j$$

We define $Q', K' \in \mathbb{R}^{n \times d}$, and let $q'_i, k'_j \in \mathbb{R}^d$, where $q'^\top_i$ and $k'^\top_j$ be the $i$-th and $j$-th row of $Q', K'$ respectively, for any $i, j \in [n]$. Let $q'_i = R^{(i)}q_i$ and $k'_j = R^{(j)}k_j$. By Equation (34) in Su et al. (2024), we know that we can get $Q', K'$ in $O(nd)$ time. Thus, we can apply $Q', K'$ in our Theorem 3.4 and Theorem 5.5 to get the same approximation error guarantee and the same time complexity.

**Extend to full self-attention.**   We can easily extend our method to full self-attention. Our proposed approach can be extended to accelerate full self-attention as well, not just the causal attention mechanism. Note that the full self-attention matrix can be split into a lower triangular matrix $L$ and an upper triangular matrix $U$. Then, our conv-basis approximation method can be applied separately to $L$ and the transpose of $U$. This allows the algorithm to handle both the lower and upper triangular components of the full attention matrix. The diagonal normalization step $D^{-1}$ would need to be adjusted to account for the full matrix rather than just the lower triangular portion. Finally, we combine the approximations of $L$ and $U^\top$ to reconstruct the full self-attention output.

**Memory consumption.**   Our method does not increase the memory consumption because each convolution matrix can be stored as a $n$-dimention vector (see Definition 2.5). Therefore, our method requires $O(kn)$ memory for $k$ convolution matrices, $O(nd)$ memory for the value matrix $V \in \mathbb{R}^{n \times d}$, and $O(n)$ memory for the diagonal matrix $D \in \mathbb{R}^{n \times n}$. In total, our memory consumption is $O(kn + nd)$. For the standard attention computation of $D^{-1}AV$, it requires $O(n^2)$ memory for the attention matrix A, $O(nd)$ memory for the value matrix $V \in \mathbb{R}^{n \times d}$, and $O(n)$ memory for the diagonal matrix $D \in \mathbb{R}^{n \times n}$. In total, the memory consumption is $O(n^2 + nd)$.

**Limitation.**   Although in this paper, we provide a comprehensive theoretical analysis aiming to reduce the quadratic computational cost $O(n^2)$, we do not have full empirical results or experiments conducted to validate the proposed algorithms on real-world benchmarks. With the rapid development of large language models, the size of input tokens is increasing. Therefore, it is urgent to develop more efficient algorithms to overcome the quadratic complexity and enable more efficient training of LLMs. Neither theoretical work nor experiments can be done trivially, and it will take more effort to successfully implement our novel theoretical results in practice even with more experimental results.

## B   Technical Details About conv Approximation

In Section B.1, we present the background of Toeplitz, circulant, and convolution matrices. In Section B.2, we develop more mathematical tools for studying the conv approximation. In Section B.3, we give the key lemmas we used. In Section B.4, we use these tools and lemmas to prove our main theorem for the conv approximation. In Section B.5, we analyze our case study.

### B.1   Properties of Toeplitz, Circulant, and Convolution Matrices

**Remark B.1.** *The integer $i$ may have different ranges. We will specify these ranges in later text, corresponding to different contexts.*

The Toeplitz matrix is one such structured matrix that has constant values along its diagonals. We define it as follows:

**Definition B.2** (Toeplitz matrix)**.** *Given a length-$(2n-1)$ vector $a \in \mathbb{R}^{2n-1}$ (for convenience, we use $a_i \in \mathbb{R}$ to denote the entry of vector where $i \in \{-(n-1), -(n-2), \cdots, 0, \cdots, (n-2), (n-1)\}$), we can formulate a function* $\mathsf{Toep} : \mathbb{R}^{2n-1} \to \mathbb{R}^{n \times n}$ *as follows*

$$
\mathsf{Toep}(a) = \begin{bmatrix} a_0 & a_{-1} & a_{-2} & \cdots & a_{-(n-1)} \\ a_1 & a_0 & a_{-1} & \cdots & a_{-(n-2)} \\ a_2 & a_1 & a_0 & \cdots & a_{-(n-3)} \\ \vdots & \vdots & \vdots & \ddots & \vdots \\ a_{n-1} & a_{n-2} & a_{n-3} & \cdots & a_0 \end{bmatrix}.
$$

Furthermore, we define the circulant matrix, which is a structured matrix where each row vector is rotated one element to the right relative to the preceding row vector, which is defined as follows:

**Definition B.3** (Circulant matrix)**.** *Let $a \in \mathbb{R}^n$ denote a length-$n$ vector. We define* $\mathsf{Circ} : \mathbb{R}^n \to \mathbb{R}^{n \times n}$ *as,*

$$
\mathsf{Circ}(a) := \begin{bmatrix} a_1 & a_n & a_{n-1} & \cdots & a_2 \\ a_2 & a_1 & a_n & \cdots & a_3 \\ a_3 & a_2 & a_1 & \cdots & a_4 \\ \vdots & \vdots & \vdots & \ddots & \vdots \\ a_n & a_{n-1} & a_{n-2} & \cdots & a_1 \end{bmatrix}.
$$

Now, we define a binary operation $*$ defined on $\mathbb{R}^d$:

**Definition B.4.** *Let* conv *be defined in Definition 2.5. Given two vectors $a$ and $x \in \mathbb{R}^n$, let $a * x \in \mathbb{R}^n$ denote the convolution operator between $a$ and $x$, i.e., $a * x := \mathsf{conv}(a)x$.*

Finally, we present a basic fact about the Hadamard product.

**Fact B.5.** *For all $a, b \in \mathbb{R}^n$, we have $a \circ b = b \circ a = \mathrm{diag}(a) \cdot b = \mathrm{diag}(b) \cdot a$.*

Below, we explore the properties of conv, Toep, Resi, and Circ.

**Claim B.6.** *Given a length-$(2n-1)$ vector $a' \in \mathbb{R}^{2n-1}$ (for convenience, we use $a_i' \in \mathbb{R}$ to denote the entry of vector where $i \in \{-(n-1), -(n-2), \cdots, 0, \cdots, (n-2), (n-1)\}$). Let $a \in \mathbb{R}^n$, such that $a = [a_0', a_1', \ldots, a_{n-1}']^\top$. Let $M$ be defined in Definition 2.2, Toep be defined in Definition B.2, and conv be defined in Definition 2.5. We have*

$$\mathrm{conv}(a) = \mathrm{Toep}(\begin{bmatrix} \mathbf{0}_{n-1} \\ a \end{bmatrix}) = M \circ \mathrm{Toep}(a').$$

*Proof.* The proof directly follows the Definition 2.2, Definition B.2, and Definition 2.5. □

**Fact B.7** (Folklore). *Let Toep be defined in Definition B.2, and Circ be defined in Definition B.3. Given a length-$(2n-1)$ vector $a \in \mathbb{R}^{2n-1}$ (for convenience, we use $a_i \in \mathbb{R}$ to denote the entry of vector where $i \in \{-(n-1), -(n-2), \cdots, 0, \cdots, (n-2), (n-1)\}$). Let $a' \in \mathbb{R}^{2n}$, such that $a' = [a_0, a_1, \ldots, a_{n-1}, 0, a_{-(n-1)}, \ldots, a_{-1}]^\top$. For any $x \in \mathbb{R}^n$, we have*

$$\mathrm{Circ}(a') \begin{bmatrix} x \\ \mathbf{0}_n \end{bmatrix} = \begin{bmatrix} \mathrm{Toep}(a) & \mathrm{Resi}(a) \\ \mathrm{Resi}(a) & \mathrm{Toep}(a) \end{bmatrix} \cdot \begin{bmatrix} x \\ \mathbf{0}_n \end{bmatrix} = \begin{bmatrix} \mathrm{Toep}(a)x \\ \mathrm{Resi}(a)x \end{bmatrix},$$

*where the residual matrix is defined as*

$$\mathrm{Resi}(a) := \begin{bmatrix} 0 & a_{n-1} & a_{n-2} & \cdots & a_2 & a_1 \\ a_{-(n-1)} & 0 & a_{n-1} & \cdots & a_3 & a_2 \\ a_{-(n-2)} & a_{-(n-1)} & 0 & \cdots & a_4 & a_3 \\ \vdots & \vdots & \vdots & \ddots & \vdots & \vdots \\ a_{-2} & a_{-3} & a_{-4} & \cdots & 0 & a_{n-1} \\ a_{-1} & a_{-2} & a_{-3} & \cdots & a_{-(n-1)} & 0 \end{bmatrix}.$$

$\mathrm{Circ}(a)$ can be expressed in the form of $F^{-1}\mathrm{diag}(Fa)F$, which is as follows:

**Fact B.8** (Folklore). *Let $a \in \mathbb{R}^n$ denote a length-$n$ vector. Let Circ be defined in Definition B.3. Let $F \in \mathbb{C}^{n \times n}$ denote the discrete Fourier transform matrix. Using the property of discrete Fourier transform, we have*

$$\mathrm{Circ}(a) = F^{-1}\mathrm{diag}(Fa)F.$$

**Claim B.9** (Restatement of Claim 2.6). *We have $\mathrm{conv}(e_j) \in \mathbb{R}^{n \times n}$ is a $j$-rank matrix, where the $j$-th entry of $e_j \in \mathbb{R}^n$ is $1$ and all other entries are $0$.*

*Proof.* This follows from Definition 2.5. □

**Claim B.10** (Restatement of Claim 2.7). *Let conv be defined in Definition 2.5. For any $a, x \in \mathbb{R}^n$, $\mathrm{conv}(a)x$ can be computed in $O(n \log n)$ via FFT.*

*Proof of Claim 2.7.* For any $a \in \mathbb{R}^n$, we denote $a' = \begin{bmatrix} \mathbf{0}_{n-1} \\ a \end{bmatrix} \in \mathbb{R}^{2n-1}$ and $a'' = \begin{bmatrix} a \\ \mathbf{0}_n \end{bmatrix} \in \mathbb{R}^{2n}$. We have

$$\begin{bmatrix} \mathrm{conv}(a)x \\ \mathrm{Resi}(a')x \end{bmatrix} = \begin{bmatrix} \mathrm{Toep}(a')x \\ \mathrm{Resi}(a')x \end{bmatrix} = \mathrm{Circ}(a'') \begin{bmatrix} x \\ \mathbf{0}_n \end{bmatrix} = F^{-1}\mathrm{diag}(Fa'')F \begin{bmatrix} x \\ \mathbf{0}_n \end{bmatrix},$$

where the first step follows Claim B.6, i.e., $\mathrm{conv}(a) = \mathrm{Toep}(\begin{bmatrix} \mathbf{0}_{n-1} \\ a \end{bmatrix})$, the second step follows Fact B.7 and the last step follows Fact B.8. We finish the proof by $O(n \log n)$ for FFT. □

**Claim B.11** (Restatement of Claim 2.8). *conv is additive, i.e., for any $a, b, x \in \mathbb{R}^n$ we have*

$$\mathrm{conv}(a)x + \mathrm{conv}(b)x = \mathrm{conv}(a+b)x.$$

*Proof.* This follows from Definition 2.5 and the fact that the matrix product operation is additive. □

**Claim B.12** (Restatement of Claim 2.10). *Let $m \in [n]$. For any $a, x \in \mathbb{R}^n$, $\mathsf{conv}(a, m)x$, (defined in Definition 2.9) can be computed in $O(n \log n)$ via FFT.*

*Proof.* This follows from considering the calculation between the truncated matrix of $\mathsf{conv}(a, m)$ and the truncated vector of $x$ with Claim 2.7. □

### B.2 MATHEMATICAL TOOLS DEVELOPMENT FOR $k$-conv BASIS

**Definition B.13.** *Let $M \in \mathbb{R}^{n \times n}$ be defined in Definition 2.2 and $Q, K \in \mathbb{R}^{n \times d}$ be defined in Definition 2.1. We define $\widetilde{H} := M \circ (QK^\top) \in \mathbb{R}^{n \times n}$.*

When a lower triangular matrix $H$ is expressed as the sum of $k$ convolution matrices, it is useful to understand the structure of the entries in $H$. The following claim provides an explicit formula for the entries of $H$ in terms of the basis vectors of the convolution matrices.

**Claim B.14.** *Given $b_1, \ldots, b_k \in \mathbb{R}^n$ and $k$ integers $m_1, m_2, \ldots, m_k$ satisfying $n \geq m_1 > m_2 > \cdots > m_k \geq 1$, let $H = \sum_{i \in [k]} \mathsf{conv}(b_i, m_i)$. Then, for any $i \geq j \in [n]$, let $\ell$ satisfy $m_\ell \geq n - j + 1$ and $m_{\ell+1} < n - j + 1$, and we have*

$$H_{i,j} = \sum_{l \in [\ell]} (b_l)_{i-j+1}.$$

*For any $i < j \in [n]$, we have $H_{i,j} = 0$.*

*Proof.* This is trivial by following $H = \sum_{i \in [k]} \mathsf{conv}(b_i, m_i)$, the Definition 2.5 and Definition 2.9. □

We present the property of $\widetilde{H} = M \circ (QK^\top)$ as follows:

**Lemma B.15.** *Given $M \in \mathbb{R}^{n \times n}$, $Q, K \in \mathbb{R}^{n \times d}$, and $\widetilde{H} = M \circ (QK^\top)$, we have for any $j \in [n]$, there exists $\widetilde{H}_j \in \mathbb{R}^n$, i.e., the $j$-th column of $\widetilde{H}$, such that*

$$\widetilde{H}_j = M_j \circ (Q(K^\top)_j)$$

*with time complexity $O(nd)$, where $(K^\top)_j$ denotes the $j$-th row of $K$.*

*Proof.* We can check the correctness as follows:

$$\begin{aligned}
(\widetilde{H})_j &= (M \circ (QK^\top))_j \\
&= M_j \circ (QK^\top)_j \\
&= M_j \circ (Q(K^\top)_j),
\end{aligned}$$

where the first step follows from the definition of $\widetilde{H}$ (see Definition B.13), the second step follows from simple algebra, the third step follows from the fact that the $j$-th column of $K^\top$ is equal to the $j$-th row of $K$.

Now, we can check the running time.

- As $Q \in \mathbb{R}^{n \times d}$ and $(K^\top)_j \in \mathbb{R}^d$, we need $O(nd)$ time to get $Q(K^\top)_j$.

- For any vector $v$, we need $O(n)$ time to get $M_j \circ v$.

Thus, in total, the time complexity is $O(nd)$. □

The key idea behind our approach is to express the matrix exponential of a matrix with $k$-conv basis as the sum of $k$ sub-convolution matrices involving the basis vectors. This allows us to efficiently approximate the exponential of the attention matrix. We show how to compute the new basis vectors of the convolution matrices from the original basis vectors below.

**Lemma B.16.** *Let $M$ be a mask defined in Definition 2.2. Given $b_1, \ldots, b_k \in \mathbb{R}^n$ and $k$ integers $m_1, m_2, \ldots, m_k$ satisfying $n \geq m_1 > m_2 > \cdots > m_k \geq 1$, we let $H = \sum_{r \in [k]} \mathsf{conv}(b_r, m_r)$. We denote $\widetilde{b}_1 = \exp(b_1)$. Then, we can get $\widetilde{b}_2, \widetilde{b}_3, \ldots \widetilde{b}_k \in \mathbb{R}^n$ such that for any $r \in \{2, 3, \cdots, k\}$*

$$\widetilde{b}_r = \exp(\sum_{l \in [r]} b_l) - \exp(\sum_{l \in [r-1]} b_l)$$

*and $M \circ \exp(H) = \sum_{r \in [k]} \mathsf{conv}(\widetilde{b}_r, m_r)$ with time complexity $O(nk)$.*

*Proof.* **Correctness.**

By Claim B.14, for any $i \geq j \in [n]$, let $\ell$ satisfy $m_\ell \geq n - j + 1$ and $m_{\ell+1} < n - j + 1$, and we have

$$H_{i,j} = \sum_{l \in [\ell]} (b_l)_{i-j+1}. \tag{1}$$

As $\exp$ is an element-wise function, when $i \geq j$ we have $(M \circ \exp(H))_{i,j} = \exp(H)_{i,j}$ and

$$
\begin{aligned}
\exp(H)_{i,j} &= \exp(\sum_{l \in [\ell]} (b_l)_{i-j+1}) \\
&= \sum_{r=1}^{\ell} \exp(\sum_{l \in [r]} (b_l)_{i-j+1}) - \exp(\sum_{l \in [r-1]} (b_l)_{i-j+1}) \\
&= \sum_{r=1}^{\ell} (\widetilde{b}_r)_{i-j+1} \\
&= \sum_{r=1}^{\ell} \mathsf{conv}(\widetilde{b}_r, m_r)_{i,j} \\
&= \sum_{r=1}^{k} \mathsf{conv}(\widetilde{b}_r, m_r)_{i,j},
\end{aligned}
$$

where the first step follows from Eq. (1), the second step follows from simple algebra, the third step follows from the lemma statement, the fourth step follows from Definition 2.9, and the last step follows from Definition 2.9 (when $k < r \leq \ell$, $\mathsf{conv}(\widetilde{b}_r, m_r)_{i,j} = 0$).

When $i < j$ we have $(M \circ \exp(H))_{i,j} = 0 = \sum_{r=1}^{k} \mathsf{conv}(\widetilde{b}_r, m_r)_{i,j}$.

Thus, we have $M \circ \exp(H) = \sum_{r \in [k]} \mathsf{conv}(\widetilde{b}_r, m_r)$.

**Running time.**

We need $O(nk)$ time to get $\sum_{l \in [r]} b_l$ for any $r \in [k]$. Then, we need $O(1)$ time for element-wise $\exp$ and minus operation for $O(nk)$ terms. Thus, in total, we need $O(nk)$ time complexity. $\square$

**Lemma B.17.** *Let $G \in \mathbb{R}^{n \times n}$. Let $M \in \{0, 1\}^{n \times n}$. Let $H = M \circ G$ and $A = M \circ \exp(G)$. Then, we have*

$$A = M \circ \exp(H).$$

*Proof.* We have

$$
\begin{aligned}
A &= M \circ \exp(G) \\
&= M \circ \exp(M \circ G) \\
&= M \circ \exp(H),
\end{aligned}
$$

where the first step follows the lemma statement, the second step follows the property of Hadamard product and the last step follows the lemma statement. $\square$

**Theorem B.18** (Restatement of Theorem 3.3). *For any lower triangular matrix $G \neq \mathbf{0}_{n \times n} \in \mathbb{R}^{n \times n}$, there exists $k, T \in [n]$ and $\delta, \epsilon \geq 0$ such that $G$ is a $\epsilon$-close $(T, \delta)$-non-degenerate $k$-conv basis matrix.*

*Proof.* By Lemma 2.12, we have $G$ is a matrix with $k$-conv basis for some $k \in [n]$. We finish the proof by setting $T = 1$ and $\delta = \epsilon = 0$. □

### B.3 LEMMA USED IN MAIN THEOREM PROOF

In this section, we present the formal proof for our conv approximation main result. In Algorithm 2, we recover the $k$-conv basis vectors $b'_1, \ldots, b'_k \in \mathbb{R}^n$ through an iterative process. We show that after each iteration $i$, the algorithm maintains certain invariants related to the recovered basis vectors $b'_1, \ldots, b'_i \in \mathbb{R}^n$, the index $s$, and the error compared to the true basis vectors $b_1, \ldots, b_i \in \mathbb{R}^n$. These properties allow us to prove the correctness of the overall algorithm. The following lemma formalizes these invariants:

**Lemma B.19.** *Let $\widetilde{H}$ be a $\epsilon$-close $(T, \delta)$-non-degenerate $k$-conv basis matrix as defined in Definition 3.2, where $\delta, \epsilon \geq 0$ and $k, T \in [n]$. Let $Q, K, V \in \mathbb{R}^{n \times d}$. In Algorithm 2, we can get $b'_1, \ldots, b'_k \in \mathbb{R}^n$. Then, for any $i \in [k]$, after the $i$-th loop, we have*

- *Part 1: $v = \sum_{r \in [i]} (b'_r)_{1:T}$ and $u = \sum_{r \in [i]} b'_r$*

- *Part 2: $s = n - m_i + 1$*

- *Part 3: $\| \sum_{r \in [i]} (b'_r)_{1:T} - \sum_{r \in [i]} (b_r)_{1:T} \|_1 \leq T\epsilon$*

- *Part 4: $| \sum_{r \in [i]} (b'_r)_l - \sum_{r \in [i]} (b_r)_l | \leq \epsilon$ for any $l \in [n]$.*

*Proof.* We use the math induction to prove the correctness.

Let $b'_1, \ldots, b'_k \in \mathbb{R}^n$ and $v \in \mathbb{R}^T$ defined in Algorithm 2. Let $i \in \{0, \ldots, k-1\}$ be fixed. Suppose after the $i$-th loop, we have

- Part 1: $v = \sum_{r \in [i]} (b'_r)_{1:T}$ and $u = \sum_{r \in [i]} b'_r$

- Part 2: $s = n - m_i + 1$ (Denote $s = 0$, after the 0-th loop.)

- Part 3: $\| \sum_{r \in [i]} (b'_r)_{1:T} - \sum_{r \in [i]} (b_r)_{1:T} \|_1 \leq T\epsilon$

- Part 4: $| \sum_{r \in [i]} (b'_r)_l - \sum_{r \in [i]} (b_r)_l | \leq \epsilon$ for any $l \in [n]$

Now we consider after the $i + 1$-th loop.

**Proof of Part 1.**

We have $v = \sum_{r \in [i+1]} (b'_r)_{1:T}$ and $u = \sum_{r \in [i+1]} b'_r$ by the line 9 and line 10 in Algorithm 2.

**Proof of Part 2.**

We denote the output of $\mathrm{SEARCH}(Q, K, k, T, \delta, \epsilon, \sum_{r \in [i]} (b'_r)_{1:T}, m_i, n - T + 1)$ as $y$. Now, we prove $y = n - m_{i+1} + 1$.

It is clear that $n - m_i + 1 \leq y \leq n - T + 1$. For any $j \in \{n - m_i + 1, \ldots, n - T + 1\}$, we have line 7 in Algorithm 3 as

$$
\begin{aligned}
\alpha &= \| (\widetilde{H}_j)_{j:j+T-1} - v \|_1 \\
&= \| (H_j)_{j:j+T-1} + R_{j,j:j+T-1} - v \|_1 \\
&= \| (H_j)_{j:j+T-1} + R_{j,j:j+T-1} - \sum_{r \in [i]} (b'_r)_{1:T} \|_1 \\
&= \| (\sum_{r \in [k]} \mathrm{conv}(b_r, m_r))_{j,j:j+T-1} + R_{j,j:j+T-1} - \sum_{r \in [i]} (b'_r)_{1:T} \|_1,
\end{aligned}
\tag{2}
$$

where the first step follows from Definition B.13 ($\widetilde{H} = H + R$), the second step follows from Part 1, and the last step follows from Definition 3.2 ($H = \sum_{r \in [k]} \mathsf{conv}(b_r, m_r)$).

When $j < n - m_{i+1} + 1$, we have Eq. (2) as

$$\|(\sum_{r \in [k]} \mathsf{conv}(b_r, m_r))_{j,j:j+T-1} + R_{j,j:j+T-1} - \sum_{r \in [i]} (b'_r)_{1:T}\|_1$$

$$\leq \|(\sum_{r \in [k]} \mathsf{conv}(b_r, m_r))_{j,j:j+T-1} - \sum_{r \in [i]} (b'_r)_{1:T}\|_1 + \|R_{j,j:j+T-1}\|_1$$

$$\leq \|(\sum_{r \in [k]} \mathsf{conv}(b_r, m_r))_{j,j:j+T-1} - \sum_{r \in [i]} (b'_r)_{1:T}\|_1 + T\epsilon$$

$$= \|(\sum_{r \in [i]} \mathsf{conv}(b_r, m_r))_{j,j:j+T-1} - \sum_{r \in [i]} (b'_r)_{1:T}\|_1 + T\epsilon$$

$$= \|\sum_{r \in [i]} (b_r)_{1:T} - \sum_{r \in [i]} (b'_r)_{1:T}\|_1 + T\epsilon$$

$$\leq 2T\epsilon$$

$$< \delta - 2T\epsilon,$$

where the first step follows from the triangle inequality, the second step follows from Definition 3.2 ($\|R\|_\infty \leq \epsilon$), the third step follows from $j < n - m_{i+1} + 1$, the fourth step follows from Definition 2.9, the fifth step follows from Part 3, and the last step follows from Definition 3.2 ($\epsilon \leq \frac{\delta}{5T} < \frac{\delta}{4T}$).

Similarly, when $j \geq n - m_{i+1} + 1$, we have Eq. (2) as

$$\|(\sum_{r \in [k]} \mathsf{conv}(b_r, m_r))_{j,j:j+T-1} + R_{j,j:j+T-1} - \sum_{r \in [i]} (b'_r)_{1:T}\|_1$$

$$\geq \|(\sum_{r \in [k]} \mathsf{conv}(b_r, m_r))_{j,j:j+T-1} - \sum_{r \in [i]} (b'_r)_{1:T}\|_1 - \|R_{j,j:j+T-1}\|_1$$

$$\geq \|(\sum_{r \in [k]} \mathsf{conv}(b_r, m_r))_{j,j:j+T-1} - \sum_{r \in [i]} (b'_r)_{1:T}\|_1 - T\epsilon$$

$$= \|(\sum_{r \in [k]} \mathsf{conv}(b_r, m_r))_{j,j:j+T-1} - \sum_{r \in [i]} (b_r)_{1:T} + \sum_{r \in [i]} (b_r)_{1:T} - \sum_{r \in [i]} (b'_r)_{1:T}\|_1 - T\epsilon$$

$$\geq \|(\sum_{r \in [k]} \mathsf{conv}(b_r, m_r))_{j,j:j+T-1} - \sum_{r \in [i]} (b_r)_{1:T}\|_1 - \|\sum_{r \in [i]} (b_r)_{1:T} - \sum_{r \in [i]} (b'_r)_{1:T}\|_1 - T\epsilon$$

$$\geq \|(\sum_{r \in [k]} \mathsf{conv}(b_r, m_r))_{j,j:j+T-1} - \sum_{r \in [i]} (b_r)_{1:T}\|_1 - 2T\epsilon$$

$$\geq \delta - 2T\epsilon$$

where the first step follows from the triangle inequality, the second step follows from Definition 3.2 ($\|R\|_\infty \leq \epsilon$), the third step follows from simple algebra, the fourth step follows from the triangle inequality, the fifth step follows from Part 3, and the last step follows from Definition 3.1.

Thus, we can claim, when $\alpha < \delta - 2T\epsilon$, we have $j < n - m_{i+1} + 1$, and we have $j \geq n - m_{i+1} + 1$ otherwise. Therefore, by binary search, we can get $s = y = n - m_{i+1} + 1$.

**Proof of Part 3.**

We have $s = n - m_{i+1} + 1$ and $u = \sum_{r \in [i]} b'_r$ at line 8 in Algorithm 2. Thus, we have

$$\|\sum_{r \in [i+1]} (b'_r)_{1:T} - \sum_{r \in [i+1]} (b_r)_{1:T}\|_1$$

$$= \|(b'_{i+1})_{1:T} + \sum_{r \in [i]} (b'_r)_{1:T} - \sum_{r \in [i+1]} (b_r)_{1:T}\|_1$$

$$= \|\widetilde{H}_{s,s:s+T-1} - u_{1:T} + \sum_{r \in [i]} (b'_r)_{1:T} - \sum_{r \in [i+1]} (b_r)_{1:T}\|_1$$

$$= \|\widetilde{H}_{s,s:s+T-1} - \sum_{r \in [i+1]} (b_r)_{1:T}\|_1$$

$$= \|H_{s,s:s+T-1} + R_{s,s:s+T-1} - \sum_{r \in [i+1]} (b_r)_{1:T}\|_1$$

$$= \| \sum_{r \in [k]} \mathsf{conv}(b_r, m_r)_{s,s:s+T-1} + R_{s,s:s+T-1} - \sum_{r \in [i+1]} (b_r)_{1:T}\|_1$$

$$= \| \sum_{r \in [i+1]} \mathsf{conv}(b_r, m_r)_{s,s:s+T-1} + R_{s,s:s+T-1} - \sum_{r \in [i+1]} (b_r)_{1:T}\|_1$$

$$= \| \sum_{r \in [i+1]} (b_r)_{1:T} + R_{s,s:s+T-1} - \sum_{r \in [i+1]} (b_r)_{1:T}\|_1$$

$$= \|R_{s,s:s+T-1}\|_1$$

$$\leq T\epsilon,$$

where the first step follows from simple algebra, the second step follows from Algorithm 2 (line 8), the third step follows from $u = \sum_{r \in [i]} b'_r$, the fourth step follows from Definition B.13 ($\widetilde{H} = H + R$), the fifth step follows from Definition 3.2 ($H = \sum_{r \in [k]} \mathsf{conv}(b_r, m_r)$), the sixth step follows from $s = n - m_{i+1} + 1$, the seventh step follows from Definition 2.9, the eighth step follows from simple algebra, and the last step follows from Definition 3.2 ($\|R\|_\infty \leq \epsilon$).

**Proof of Part 4.**

We can get $|\sum_{r \in [i+1]} (b'_r)_l - \sum_{r \in [i]} (b_r)_l| \leq \epsilon$ for any $l \in [n]$ similarly as Proof of Part 3.

We can check the initial conditions hold. Thus, we finish the whole proof by math induction. $\square$

Building upon Lemma B.19, we now analyze the overall error of our approach for approximating the attention computation. Recall that our goal is to efficiently approximate the matrix $Y = D^{-1}AV$, where $A = M \circ \exp(QK^\top)$ and $D = \mathrm{diag}(A\mathbf{1}_n)$. We will show that by using the approximate basis vectors recovered by Algorithm 2, we can construct matrices $\widetilde{A}$ and $\widetilde{D}$ such that the approximation error $\|Y - \widetilde{D}^{-1}\widetilde{A}V\|_\infty$ is bounded. The following lemma provides this error analysis:

**Lemma B.20** (Error analysis). *Let $\widetilde{H}$ be a $\epsilon$-close $(T, \delta)$-non-degenerate $k$-conv basis matrix as defined in Definition 3.2, where $\delta, \epsilon \geq 0$ and $k, T \in [n]$. Let $Q, K, V \in \mathbb{R}^{n \times d}$. Recall $A = M \circ \exp(QK^\top)$ and $D = \mathrm{diag}(A\mathbf{1}_n)$ defined in Definition 2.3. By Algorithm 2, we can get $k$-conv basis $\widetilde{b}_1, \ldots, \widetilde{b}_k \in \mathbb{R}^n$ and $k$ integers $m_1, m_2, \ldots, m_k$ satisfying $n \geq m_1 > m_2 > \cdots > m_k \geq T$, such that $\widetilde{A} := \sum_{r \in [k]} \mathsf{conv}(\widetilde{b}_r, m_r)$ and $\widetilde{D} := \mathrm{diag}(\widetilde{A}\mathbf{1}_n)$ satisfy*

$$\|D^{-1}AV - \widetilde{D}^{-1}\widetilde{A}V\|_\infty \leq 2(\exp(2\epsilon) - 1)\|V\|_\infty,$$

*with time complexity $O(knd \log(n))$.*

*Proof.* **Correctness.**

By Lemma B.19 Part 4, we can get $b'_1, \ldots, b'_k \in \mathbb{R}^n$, such that, for any $i \in [k]$ and $l \in [n]$, we have

$$|\sum_{r \in [i]} (b'_r)_l - \sum_{r \in [i]} (b_r)_l| \leq \epsilon. \tag{3}$$

Furthermore, we denote

$$H' = \sum_{r \in [k]} \mathsf{conv}(b'_r, m_r)$$

Recall $\widetilde{H} = H + R \in \mathbb{R}^{n \times n}$,

$$H = \sum_{r \in [k]} \mathsf{conv}(b_r, m_r),$$

and $\|R\|_\infty \le \epsilon$.

Thus, we have

$$\begin{aligned}
\|H' - \widetilde{H}\|_\infty &\le \|H' - H\|_\infty + \|H - \widetilde{H}\|_\infty \\
&\le \|H' - H\|_\infty + \|R\|_\infty \\
&\le 2\epsilon,
\end{aligned} \tag{4}$$

where the first step follows from triangle inequality, the second step follows from $\widetilde{H} = H + R$ and the last step follows from $\|R\|_\infty \le \epsilon$ and Eq. (3).

By Lemma B.17, we have

$$\begin{aligned}
A &= M \circ \exp(QK^\top) \\
&= M \circ \exp(M \circ QK^\top) \\
&= M \circ \exp(\widetilde{H}).
\end{aligned}$$

We also have

$$\widetilde{A} = \sum_{r \in [k]} \mathsf{conv}(\widetilde{b}_r, m_r) = M \circ \exp(H')$$

by Lemma B.16 and line 12 in Algorithm 2.

Then, by Lemma E.4, we have

$$\|D^{-1}AV - \widetilde{D}^{-1}\widetilde{A}V\|_\infty \le 2(\exp(2\epsilon) - 1)\|V\|_\infty.$$

**Running time.**

We have $k$ loops in Algorithm 2.

In each loop, we call $O(\log(n))$ times of binary search function. In each binary search function, we take $O(nd)$ time for line 6 in Algorithm 3 by Lemma B.15. Thus, we take $O(nd\log(n))$ in total for the search (Algorithm 3) in each loop.

In each loop, we take $O(nd)$ time for line 7 in Algorithm 2 by Lemma B.15.

Thus, we take total $O(k(nd + nd\log(n))) = O(knd\log(n))$ for the whole loop.

We take $O(nk)$ time for the line 12 in Algorithm 2 by Lemma B.16.

In total, we take $O(nk + knd\log(n)) = O(knd\log(n))$ time. □

We are now ready to prove our main result for the conv approximation approach. Theorem B.21 brings together the key components we have developed: the existence of a $k$-conv basis for the attention matrix (Definition 3.2), the ability to efficiently recover an approximate $k$-conv basis (Algorithm 2 and Lemma B.19), and the bounded approximation error when using this approximate basis (Lemma B.20). The theorem statement is a formal version of our main conv result, Theorem 3.4 and Algorithm 1, which was presented in the main text. It specifies the input properties, the approximation guarantees, and the time complexity of our approach.

## B.4 PROOF OF MAIN THEOREM

**Theorem B.21** (Main conv results for inference (Restatement of Theorem 3.4))**.** *Let $Q, K, V \in \mathbb{R}^{n \times d}$. Recall $A = M \circ \exp(QK^\top) \in \mathbb{R}^{n \times n}$, $D = \mathrm{diag}(A\mathbf{1}_n) \in \mathbb{R}^{n \times n}$ defined in Definition 2.3. We denote $Y := D^{-1}AV \in \mathbb{R}^{n \times d}$. Let $M \circ (QK^\top)$ be a $\epsilon$-close $(T, \delta)$-non-degenerate $k$-conv basis matrix as defined in Definition 3.2, where $\delta, \epsilon \ge 0$ and $k, T \in [n]$. By Algorithm 1, we can get $\widetilde{Y}$ such that*

$$\|Y - \widetilde{Y}\|_\infty \le 2(\exp(2\epsilon) - 1)\|V\|_\infty,$$

*whose time complexity is $O(knd\log(n))$ given $M, Q, K, V$.*

*Proof of Theorem 3.4.* **Correctness.**

Correctness follows Lemma B.20.

**Running time.**

By Lemma B.20, we need time $O(knd\log(n))$ time to get $k$-conv basis $\widetilde{b}_1, \ldots, \widetilde{b}_k \in \mathbb{R}^n$ and $k$ integers $m_1, m_2, \ldots, m_k$ satisfying $n \geq m_1 > m_2 > \cdots > m_k \geq T$.

Denote $\widetilde{A} := \sum_{r \in [k]} \mathsf{conv}(\widetilde{b}_r, m_r)$. By Claim 2.10, we take $O(knd\log(n))$ time to get $\widetilde{A}V$ via FFT as $k$-conv basis and $d$ columns in $V$. Similarly, by Claim 2.10, we take $O(kn\log(n))$ time for $\widetilde{D} = \mathrm{diag}(\widetilde{A}\mathbf{1}_n)$ via FFT as $k$-conv basis. Finally, we take $O(nd)$ time to get $\widetilde{D}^{-1}\widetilde{A}V$ as $\widetilde{D}^{-1}$ is a diagonal matrix.

Thus, in total, we take $O(knd\log(n) + knd\log(n) + kn\log(n) + nd) = O(knd\log(n))$ time complexity. $\qquad\square$

**Corollary B.22** (Exact conv inference, restatement of Corollary 3.5)**.** *Let* $Q, K, V \in \mathbb{R}^{n \times d}$. *Recall* $A = M \circ \exp(QK^\top) \in \mathbb{R}^{n \times n}$, $D = \mathrm{diag}(A\mathbf{1}_n) \in \mathbb{R}^{n \times n}$ *defined in Definition 2.3. We denote* $Y := D^{-1}AV \in \mathbb{R}^{n \times d}$. *For any* $\epsilon \geq 0$ *and any* $Q, K, V$, *there exists hyper-parameter* $k, T \in [n]$ *and* $\delta \geq 0$ *such that Algorithm 1 can output* $\widetilde{Y}$ *satisfying*

$$\|Y - \widetilde{Y}\|_\infty \leq 2(\exp(2\epsilon) - 1)\|V\|_\infty.$$

*Furthermore, we can exactly get* $Y$, *i.e.,* $\epsilon = 0$, *through Algorithm 1 with time complexity* $O(n^2 d\log(n))$ *in the worst case.*

*Proof.* We set $k = n$, $T = 1$, $\delta = 0$ and $\epsilon = 0$ as the input of Algorithm 1. Then, the proof follows Theorem 3.3 and Theorem 3.4 . $\qquad\square$

### B.5 CONSTRUCTION FOR CASE STUDY

In this section, we present the case study. We use $\mathbf{i}$ to denote the $\sqrt{-1}$. For a complex number $z = a + b\mathbf{i} \in \mathbb{C}$, where $a, b \in \mathbb{R}$, we use $|z|$ to denote its norm, i.e., $|z| = \sqrt{a^2 + b^2}$.

**Lemma B.23** (Complex vector construction)**.** *If the vectors* $x_1, \cdots, x_n \in \mathbb{C}^d$ *satisfy the following properties,*

- $\|x_i\|_2 = 1$ *for all* $i \in [n]$

- *For each* $i \in [n]$, *let* $x_{i,1} = e^{\mathbf{i}i\theta}$ *and* $e_{i,l} = 0$ *for all* $l \neq 1$

*Then we have for all* $i \in [n]$, *for all* $j \in [n]$, $\|x_i - x_j\|_2^2 = f(i - j)$ *for some function* $f$.

*Proof.* We can show that

$$\begin{aligned}
\|x_i - x_j\|_2^2 &= |e^{\mathbf{i}i\theta} - e^{\mathbf{i}j\theta}|^2 \\
&= |e^{\mathbf{i}j\theta}|^2 \cdot |e^{\mathbf{i}(i-j)\theta} - 1|^2 \\
&= |e^{\mathbf{i}(i-j)\theta} - 1|^2 \\
&=: f(i - j),
\end{aligned}$$

where the first step follows from the assumption that for each $i \in [n]$ and $l \neq 1$, $x_{i,1} = e^{\mathbf{i}i\theta}$ and $e_{i,l} = 0$, the second step follows from simple algebra, the third step follows from the $|e^{\mathbf{i}j\theta}| = 1$, and the last step follows from the definition of the function $f$.

Thus, we complete the proof. $\qquad\square$

**Lemma B.24** (Real vector construction)**.** *If the vectors* $x_1, \cdots, x_n \in \mathbb{R}^d$ *satisfy the following properties,*

- $\|x_i\|_2 = 1$ *for all* $i \in [n]$

- $x_{i,1} = \cos(i\theta)$ and $x_{i,2} = \sin(i\theta)$. For all $l \notin \{1,2\}$, we have $x_{i,l} = 0$.

Then we have for all $i \in [n]$, for all $j \in [n]$, $\|x_i - x_j\|_2^2 = f(i-j)$ for some function $f$.

*Proof.* We can show that

$$
\begin{aligned}
\|x_i - x_j\|_2^2 &= (\cos(i\theta) - \cos(j\theta))^2 + (\sin(i\theta) - \sin(j\theta))^2 \\
&= 2 - 2\cos(i\theta)\cos(j\theta) - 2\sin(i\theta)\sin(j\theta) \\
&= 2 - 2\cos((i-j)\theta),
\end{aligned}
$$

where the first step follows from construction condition, the second step follows from simple algebra, and the last step follows from the trigonometric properties.

Thus, we complete the proof. $\qquad\square$

**Lemma B.25** (A general real vector construction). *If the vectors $x_1, \cdots, x_n \in \mathbb{R}^d$ satisfy the following properties,*

- $\|x_i\|_2 = 1$ *for all $i \in [n]$.*

- *Let $H \in \mathbb{R}^{d \times d}$ be any orthonormal matrix.*

- *Let $(s_1, s_2, \ldots, s_d)$ be a permutation of $(1, 2, \ldots, d)$.*

- *Let $l = \lfloor (d+1)/2 \rfloor$, where $l$ is an integer. Let $a_1, \ldots, a_l \in \mathbb{R}$.*

- *Let $u_1, \cdots, u_n \in \mathbb{R}^d$ and $x_i = Hu_i$ for any $i \in [n]$.*

- *When $d$ is even, $u_{i,s_k} = a_k \cos(i\theta_k)$ and $u_{i,s_{k+l}} = a_k \sin(i\theta_k)$, for all $k \in [l]$ and $i \in [n]$, where $\theta_1, \ldots, \theta_l \in \mathbb{R}$.*

- *When $d$ is odd, $u_{i,s_k} = a_k \cos(i\theta_k)$ and $u_{i,s_{k+l}} = a_k \sin(i\theta_k)$, for all $k \in [l-1]$ and $i \in [n]$, where $\theta_1, \ldots, \theta_{l-1} \in \mathbb{R}$, and $u_{i,s_l} = a_l$.*

*Then we have for all $i \in [n]$, for all $j \in [n]$, $\|x_i - x_j\|_2^2 = f(i-j)$ for some function $f$.*

*Proof.* When $d$ is even, we can show that

$$
\begin{aligned}
\|x_i - x_j\|_2^2 &= \|u_i - u_j\|_2^2 \\
&= \sum_{k \in [l]} (a_k \cos(i\theta_k) - a_k \cos(j\theta_k))^2 + (a_k \sin(i\theta_k) - a_k \sin(j\theta_k))^2 \\
&= \sum_{k \in [l]} a_k^2 \cos^2(i\theta_k) + a_k^2 \cos^2(j\theta_k) - 2a_k^2 \cos(i\theta_k)\cos(j\theta_k) \\
&\quad + a_k^2 \sin^2(i\theta_k) + a_k^2 \sin^2(j\theta_k) - 2a_k^2 \sin(i\theta_k)\sin(j\theta_k) \\
&= \sum_{k \in [l]} 2a_k^2 - 2a_k^2 \cos(i\theta_k)\cos(j\theta_k) - 2a_k^2 \sin(i\theta_k)\sin(j\theta_k) \\
&= \sum_{k \in [l]} 2a_k^2 (1 - \cos(i\theta_k)\cos(j\theta_k) - \sin(i\theta_k)\sin(j\theta_k)) \\
&= \sum_{k \in [l]} 2a_k^2 (1 - \cos((i-j)\theta_k)),
\end{aligned}
$$

where the first step follows $H$ being orthonormal, which preserves the Euclidean distance between two vectors, i.e., $\|Hu_1 - Hu_2\|_2 = \|u_1 - u_2\|_2$ for any $u_1, u_2 \in \mathbb{R}^d$, the second step follows from the construction condition, the third step follows from $(a-b)^2 = a^2 + b^2 - 2ab$ for all $a, b \in \mathbb{C}$, the fourth step follows from $\sin^2(x) + \cos^2(x) = 1$, the fifth step follows from simple algebra, and the last step follows from the trigonometric properties.

When $d$ is odd, we can show similar results by the same way. Thus, we complete the proof. $\qquad\square$

**Lemma B.26.** *If the following conditions hold*

- *Let $b \in \mathbb{R}^n$ denote a vector*

- *$Q \in \mathbb{R}^{n \times d}$ and $K \in \mathbb{R}^{n \times d}$*

- *For each $i, j \in [n]$,*

  - *$(QK^\top)_{i,j} = b_{i-j+1}$ if $i \geq j$*
  - *$(QK^\top)_{i,j} = b_{i-j+n+1}$ if $i < j$*

*Then, there is a vector $a = \exp(b)$ such that*

$$\exp(QK^\top) = \mathsf{Circ}(a)$$

*Proof.* Since $a = \exp(b)$, we have

$$\begin{aligned}
\mathsf{Circ}(a) &= \mathsf{Circ}(\exp(b)) \\
&= \exp(\mathsf{Circ}(b)),
\end{aligned} \tag{5}$$

where the second step follows from the fact that $\exp(\cdot)$ is applied entry-wisely to a vector.

By the assumption from the Lemma statement that $(QK^\top)_{i,j} = b_{i-j+1}$ if $i \geq j$ and $(QK^\top)_{i,j} = b_{i-j+n+1}$ if $i < j$, we get

$$QK^\top = \begin{bmatrix} b_1 & b_n & b_{n-1} & \cdots & b_2 \\ b_2 & b_1 & b_n & \cdots & b_3 \\ b_3 & b_2 & b_1 & \cdots & b_4 \\ \vdots & \vdots & \vdots & \ddots & \vdots \\ b_n & b_{n-1} & b_{n-2} & \cdots & b_1 \end{bmatrix},$$

which is exactly equal to $\mathsf{Circ}(b)$ (see Definition B.3).

Therefore, combining with Eq. (5), we have

$$\exp(QK^\top) = \mathsf{Circ}(a),$$

which completes the proof. $\square$

**Lemma B.27.** *If the following conditions hold*

- *Let $b \in \mathbb{R}^{2n-1}$ denote a vector*

- *$Q \in \mathbb{R}^{n \times d}$ and $K \in \mathbb{R}^{n \times d}$*

- *For each $i, j \in [n]$, $(QK^\top)_{i,j} = b_{i-j}$.*

*Then, there is a vector $a = \exp(b)$ such that*

$$\exp(QK^\top) = \mathsf{Toep}(a).$$

*Proof.* We can prove similarly as Lemma B.26. $\square$

**Assumption B.28.** *We assume that $W_Q W_K^\top$ is a p.s.d. matrix, so that $W_Q W_K^\top = AA^\top$ where $A \in \mathbb{R}^{d \times d}$.*

**Definition B.29.** *Assume Assumption B.28. We define $Z := XA \in \mathbb{R}^{n \times d}$, where $Z = \begin{bmatrix} z_1^\top \\ \vdots \\ z_n^\top \end{bmatrix}$. Then we have $QK^\top = ZZ^\top$.*

**Lemma B.30.** *If the following conditions hold,*

- *Assume Assumption B.28.*

- *Let $b \in \mathbb{R}^{2n-1}$ denote a vector*

- *Let $z_1, \ldots, z_n$ defined in Definition B.29 satisfy the properties in Lemma B.25.*

*Then, there is a vector $a = \exp(b)$ such that*

$$\exp(QK^\top) = \mathsf{Toep}(a).$$

*Proof.* By Lemma B.25, we have for all $i \in [n]$, for all $j \in [n]$,

$$\|z_i - z_j\|_2^2 = f(i - j)$$

for some function $f$.

We also have

$$\langle z_i, z_j \rangle = 1 - f(i - j)/2 =: g(i - j)$$

as $\|z_i\|_2 = \|z_j\|_2 = 1$.

Then, we have $\forall i, j \in [n]$,

$$
\begin{aligned}
(QK^\top)_{i,j} &= (ZZ^\top)_{i,j} \\
&= \langle z_i, z_j \rangle \\
&= g(i - j),
\end{aligned}
$$

where the first two steps from Definition B.29, and the last step from Lemma B.25. We finish the proof by denote $b_{i-j}$ as $g(i - j)$ in Lemma B.27. $\qquad\square$

## C   conv APPROXIMATION IN GRADIENT

In Section C.1, we present the basic definitions. In Section C.2, we combine all these definitions to form the loss function. In Section C.3, we analyze the running time. In Section C.4, we present the proof of the main theorem of conv approximation in gradient.

### C.1   DEFINITIONS

In this section, we let $x, y \in \mathbb{R}^{d^2}$ denote the vectorization of $X, Y \in \mathbb{R}^{d \times d}$. To concisely express the loss function, we define more functions below.

**Definition C.1.** *Let $u(x)_{j_0} \in \mathbb{R}$ (see Definition 4.5). For each $j_0 \in [n]$, we define $\alpha(x)_{j_0} : \mathbb{R}^{d^2} \to \mathbb{R}$*

$$\alpha(x)_{j_0} := \langle \underbrace{u(x)_{j_0}}_{n \times 1}, \underbrace{\mathbf{1}_n}_{n \times 1} \rangle.$$

*Consider $\alpha(x) \in \mathbb{R}^n$ as a vector whose $j_0$-th entry equals $\alpha(x)_{j_0}$.*

**Definition C.2.** *Let $\alpha(x)_{j_0} \in \mathbb{R}$ (see Definition C.1). Let $u(x)_{j_0} \in \mathbb{R}^n$ (see Definition 4.5). For a fixed $j_0 \in [n]$, we define $f(x)_{j_0} : \mathbb{R}^{d^2} \to \mathbb{R}^n$*

$$f(x)_{j_0} := \underbrace{\alpha(x)_{j_0}^{-1}}_{\text{scalar}} \underbrace{u(x)_{j_0}}_{n \times 1}.$$

*Consider $f(x) \in \mathbb{R}^{n \times n}$ as a matrix whose $j_0$-th row equals $(f(x)_{j_0})^\top$.*

**Definition C.3.** *For a fixed $i_0 \in [d]$, define $h(x)_{i_0} : \mathbb{R}^{d^2} \to \mathbb{R}^n$:*

$$h(y)_{i_0} := \underbrace{A_3}_{n \times d} \underbrace{Y_{*,i_0}}_{d \times 1},$$

*where $Y \in \mathbb{R}^{d \times d}$ is the matrix representation of $y \in \mathbb{R}^{d^2}$. Let $h(y) \in \mathbb{R}^{n \times d}$ be a matrix where $i_0$ column is $h(y)_{i_0}$.*

## C.2 Loss Functions

Now, we start the construction of the loss function.

**Definition C.4.** *For each $j_0 \in [n]$, we denote the normalized vector defined by Definition C.2 as $f(x)_{j_0} \in \mathbb{R}^n$. Similarly, for each $i_0 \in [d]$, we define $h(y)_{i_0}$ as specified in Definition C.3.*

*Consider every $j_0 \in [n]$, every $i_0 \in [d]$. Let us consider $c(x)_{j_0,i_0} : \mathbb{R}^{d^2} \times \mathbb{R}^{d^2} \to \mathbb{R}$ as follows:*

$$c(x)_{j_0,i_0} := \langle f(x)_{j_0}, h(y)_{i_0} \rangle - E_{j_0,i_0}.$$

*Here $E_{j_0,i_0}$ is the $(j_0, i_0)$-th entry of $E \in \mathbb{R}^{n \times d}$ with $j_0 \in [n], i_0 \in [d]$, similar for $\underbrace{c(x)}_{n \times d} =$*

$$\underbrace{f(x)}_{n \times n} \underbrace{h(y)}_{n \times d} - \underbrace{E}_{n \times d}.$$

**Definition C.5.** *For every $j_0 \in [n]$, for every $i_0 \in [d]$, we define $L(x)_{j_0,i_0}$ to be $:= 0.5 c(x)^2_{j_0,i_0}$.*

**Definition C.6.** *Consider $c(x) \in \mathbb{R}^{n \times d}$ which is described in Definition C.4, and $h(y) \in \mathbb{R}^{n \times d}$ which is defined in Definition C.3. We now define $q(x) \in \mathbb{R}^{n \times n}$*

$$q(x) := \underbrace{c(x)}_{n \times d} \underbrace{h(y)^\top}_{d \times n}$$

*Subsequently, we denote the $j_0$-th row of $q(x) \in \mathbb{R}^{n \times n}$ as $q(x)^\top_{j_0}$.*

**Definition C.7.** *Let $j_0 \in [n]$. We define $p(x)_{j_0} : \mathbb{R}^{d^2} \to \mathbb{R}^n$*

$$p(x)_{j_0} := (\mathrm{diag}(f(x)_{j_0}) - f(x)_{j_0} f(x)^\top_{j_0}) q(x)_{j_0}$$
$$= p_1(x)_{j_0} + p_2(x)_{j_0},$$

*where*

$$p_1(x)_{j_0} := \mathrm{diag}(f(x)_{j_0}) q(x)_{j_0}$$
$$p_2(x)_{j_0} := f(x)_{j_0} f(x)^\top_{j_0} q(x)_{j_0}.$$

*We establish $p(x) \in \mathbb{R}^{n \times n}$ such that $p(x)^\top_{j_0}$ represents the $j_0$-th row of $p(x)$. Note that $p_1(x) = f(x) \circ q(x)$.*

**Lemma C.8.** *Let $M \in \mathbb{R}^{n \times n}$ be a casual attention mask defined in Definition 2.2. Let $X \in \mathbb{R}^{n \times n}$, we have*

$$\frac{\mathrm{d}(M \circ X)}{\mathrm{d}X_{i,j}} = M \circ \frac{\mathrm{d}X}{\mathrm{d}X_{i,j}}.$$

*Proof.* The proof is trivial by element-wise multiplication. $\square$

**Lemma C.9** (Gradient computation). *We have $f(x) \in \mathbb{R}^{n \times n}$, $c(x) \in \mathbb{R}^{n \times d}$, $h(y) \in \mathbb{R}^{n \times d}$, $q(x) \in \mathbb{R}^{n \times n}$, and $p(x) \in \mathbb{R}^{n \times n}$ respectively be defined in Definitions C.2, C.4, C.3, C.6, and C.7. Consider $A_1, A_2 \in \mathbb{R}^{n \times d}$ as given and $\mathsf{A} = A_1 \otimes A_2$. We have $L(x)$ be specified in Definition 4.1, and $L(x)_{j_0,i_0}$ is as in Definition C.5.*

*Then, we can show that $\frac{\mathrm{d}L(x)}{\mathrm{d}x} = \mathrm{vec}(A_1^\top p(x) A_2)$.*

*Proof.* From the Lemma statement, by Lemma C.8, we have

$$\frac{\mathrm{d}L(x,y)_{j_0,i_0}}{\mathrm{d}x_i} = c(x,y)_{j_0,i_0} \cdot (\langle M_{j_0,*} \circ f(x)_{j_0} \circ \mathsf{A}_{j_0,i}, h(y)_{i_0} \rangle - \langle f(x)_{j_0}, h(y)_{i_0} \rangle \cdot \langle M_{j_0,*} \circ f(x)_{j_0}, \mathsf{A}_{j_0,i} \rangle)$$

$$= c(x,y)_{j_0,i_0} \cdot (\langle f(x)_{j_0} \circ \mathsf{A}_{j_0,i}, h(y)_{i_0} \rangle - \langle f(x)_{j_0}, h(y)_{i_0} \rangle \cdot \langle f(x)_{j_0}, \mathsf{A}_{j_0,i} \rangle), \quad (6)$$

where the first step is from the chain rule and the second step follows from $M_{j_0,*} \circ f(x)_{j_0} = f(x)_{j_0}$.

Note that by Fact B.5, it holds that

$$\langle f(x)_{j_0} \circ \mathsf{A}_{j_0,i}, h(y)_{i_0} \rangle = \mathsf{A}^\top_{j_0,i} \mathrm{diag}(f(x)_{j_0}) h(y)_{i_0}$$

and

$$\langle f(x)_{j_0}, v \rangle \cdot \langle f(x)_{j_0}, \mathsf{A}_{j_0,i} \rangle = \mathsf{A}_{j_0,i}^\top f(x)_{j_0} f(x)_{j_0}^\top h(y)_{i_0}$$

Therefore, Eq. (6) becomes

$$\frac{\mathrm{d}L(x)_{j_0,i_0}}{\mathrm{d}x_i} = c(x,y)_{j_0,i_0} \cdot (\mathsf{A}_{j_0,i}^\top \operatorname{diag}(f(x)_{j_0})h(y)_{i_0} - \mathsf{A}_{j_0,i}^\top f(x)_{j_0} f(x)_{j_0}^\top h(y)_{i_0})$$

$$= c(x,y)_{j_0,i_0} \cdot \mathsf{A}_{j_0,i}^\top (\operatorname{diag}(f(x)_{j_0}) - f(x)_{j_0} f(x)_{j_0}^\top)h(y)_{i_0}, \tag{7}$$

where the last step is by simple algebra.

Let $q(x)_{j_0}$ be defined as in Definition C.6:

$$q(x)_{j_0} := \sum_{i_0=1}^{d} c(x)_{j_0,i_0} h(y)_{i_0}. \tag{8}$$

Let $p(x)_{j_0}$ be define as in Definition C.7:

$$p(x)_{j_0} := (\operatorname{diag}(f(x)_{j_0}) - f(x)_{j_0} f(x)_{j_0}^\top)q(x)_{j_0}. \tag{9}$$

It holds that

$$\frac{\mathrm{d}L(x)}{\mathrm{d}x}$$

$$= \sum_{j_0=1}^{n} \sum_{i_0=1}^{d} \frac{\mathrm{d}L(x)_{j_0,i_0}}{\mathrm{d}x}$$

$$= \sum_{j_0=1}^{n} \sum_{i_0=1}^{d} \underbrace{c(x)_{j_0,i_0}}_{\text{scalar}} \cdot \underbrace{\mathsf{A}_{j_0}^\top}_{d^2 \times n} \underbrace{(\operatorname{diag}(f(x)_{j_0}) - f(x)_{j_0} f(x)_{j_0}^\top)}_{n \times n} \underbrace{h(y)_{i_0}}_{n \times 1}$$

$$= \sum_{j_0=1}^{n} \mathsf{A}_{j_0}^\top (\operatorname{diag}(f(x)_{j_0}) - f(x)_{j_0} f(x)_{j_0}^\top) q(x)_{j_0}$$

$$= \sum_{j_0=1}^{n} \mathsf{A}_{j_0}^\top p(x)_{j_0}$$

$$= \operatorname{vec}(\underbrace{A_1^\top}_{d \times n} \underbrace{p(x)}_{n \times n} \underbrace{A_2}_{n \times d})$$

where the 1st step is because of Definition 4.1, the second step follows from Eq. (7), the third step follows from Eq. (8), the fourth step follows from Eq. (9), and the fifth step follows from Fact E.9. □

## C.3 RUNNING TIME

In this section, we analyze the running time of the conv approximation approach for computing the training forward pass and backward gradient. We build upon the key definitions and loss functions introduced in the previous sections to derive the running time of the algorithm.

**Lemma C.10.** *If we have*

- *Define $u(x) \in \mathbb{R}^{n \times n}$ as outlined in Definition 4.5.*

- *Define $f(x) \in \mathbb{R}^{n \times n}$ as specified in Definition C.2.*

- *Define $h(y) \in \mathbb{R}^{n \times d}$ according to Definition C.3.*

- *Suppose $u(x)$ is a $k$-conv matrix defined in Definition 2.11 with known basis.*

*Then, we have*

- *For any $w \in \mathbb{R}^n$, we have $f(x) \cdot w \in \mathbb{R}^n$ can be done in $O(kn \log n)$ time.*

- *$h(y)$ can be expiciltiy computed in $\mathcal{T}_{\mathrm{mat}}(n, d, d)$ time.*

*Proof.* For the first part, by definition of $u(x) \in \mathbb{R}^{n \times n}$, we know that for any vector $w \in \mathbb{R}^n$, we can compute $u(x)w$ in $O(kn \log n)$ time (Claim 2.10). Thus,

$$
\begin{aligned}
f(x) \cdot w &= \mathrm{diag}(\alpha(x))^{-1} u(x) w \\
&= \mathrm{diag}(u(x)\mathbf{1}_n)^{-1} u(x) w,
\end{aligned}
$$

which can be done in $O(kn \log n)$ time by Fact B.5.

The second part is trivial by Definition C.3. $\qquad\square$

**Lemma C.11.** *If we have*

- *Define $f(x) \in \mathbb{R}^{n \times n}$ as specified in Definition C.2.*

- *Define $h(y) \in \mathbb{R}^{n \times d}$ according to Definition C.3 and $h(y)$ is known.*

- *Define $c(x) \in \mathbb{R}^{n \times d}$ as outlined in Definition C.4.*

- *Suppose $f(x)w$ takes $O(kn \log n)$ time.*

*Then, we can show that*

- *$c(x)$ can be expiciltiy computed in $O(knd \log n)$ time.*

*Proof.* Firstly we can compute $f(x)h(y)$, this can be done in $O(knd \log n)$, since we run $f(x)$ times a vector oracle (Lemma C.10) for $d$ times.

Then do minus $E \in \mathbb{R}^{n \times d}$ matrix. This takes $O(nd)$ time. Thus we complete the proof. $\qquad\square$

**Lemma C.12.** *If the following conditions hold*

- *Let $c(x) \in \mathbb{R}^{n \times d}$ be defined in Definition C.4 and $c(x)$ is known.*

- *Let $h(y) \in \mathbb{R}^{n \times d}$ be defined in Definition C.3 and $h(y)$ is known.*

- *Let $q(x) \in \mathbb{R}^{n \times n}$ be defined in Definition C.6.*

*Then, we can show that*

- *$q(x)$'s rank-d factorization can be explilcitly computed in $O(nd)$ time.*

*Proof.* Note that $q(x) = c(x)h(y)^\top$. Since both $c(x)$ and $h(y)$ are known. Thus, the result is trivial. $\qquad\square$

**Lemma C.13** (Fast computation $p_1(x)$ multiply with a vector ). *If the following conditions hold*

- *Let $f(x) \in \mathbb{R}^{n \times n}$ be defined in Definition C.2.*

- *Suppose $f(x)w$ can be done in $O(kn \log n)$ time for any $w \in \mathbb{R}^n$.*

- *Let $q(x)$ denote a rank-$\tau$ matrix with known low-rank factorizations.*

- *Let $p_1(x) = f(x) \circ q(x)$.*

*Then, we can show*

- *For any vector $w \in \mathbb{R}^n$, $p_1(x) \cdot w$ can be computed in $O(\tau kn \log n)$ time*

*Proof.* Since $q(x) \in \mathbb{R}^{n \times n}$ has rank-$\tau$, we assume that the low-rank factors are $a_1, a_2, \cdots, a_\tau \in \mathbb{R}^n$ and $b_1, b_2, \cdots, b_\tau \in \mathbb{R}^n$. In particular, $q(x)$ can be written as

$$q(x) = \sum_{i=1}^{\tau} a_i b_i^\top$$

Using a standard linear algebra trick, we can show that

$$f(x) \circ q(x) = (f(x)) \circ \left(\sum_{i=1}^{\tau} a_i b_i^\top\right)$$

$$= \sum_{i=1}^{\tau} (f(x)) \circ (a_i b_i^\top)$$

$$= \sum_{i=1}^{\tau} \mathrm{diag}(a_i) f(x) \mathrm{diag}(b_i)$$

Note that for each $i \in [\tau]$, we can show that $\mathrm{diag}(a_i) f(x) \mathrm{diag}(b_i) w$ can be computed in $O(kn \log n)$ time by Lemma statement. Thus, for any vector $w \in \mathbb{R}^n$, $(f(x) \circ q(x)) \cdot w$ can be computed in $O(\tau kn \log n)$ time. Therefore, we complete the proof.

$\square$

**Lemma C.14** (Fast computation for $r(x)$). *If the following conditions hold*

- *Let $r(x)_{j_0} := \langle f(x)_{j_0}, q(x)_{j_0} \rangle$.*

- *Let $f(x) \in \mathbb{R}^{n \times n}$ be defined in Definition C.2.*

- *Suppose $f(x)w$ can be done in $O(kn \log n)$ time for any $w \in \mathbb{R}^n$.*

- *Let $q(x)$ denote a rank-$\tau$ matrix with known low-rank factorizations.*

*Then, we can show*

- *$r(x) \in \mathbb{R}^n$ can be in $O(\tau kn \log n)$ time.*

*Proof.* Since $q(x) \in \mathbb{R}^{n \times n}$ has rank-$\tau$, we assume that the low-rank factors are $a_1, a_2, \cdots, a_\tau \in \mathbb{R}^n$ and $b_1, b_2, \cdots, b_\tau \in \mathbb{R}^n$, in particular, $q(x)$ can be written as

$$q(x) = \sum_{i=1}^{\tau} a_i b_i^\top$$

Let $q(x) = U_a U_b^\top$. It is easy to see that $f(x) q(x)^\top$ can be written as $f(x) U_b U_a^\top$.

We firstly compute $f(x) U_b$, since $U_b$ has $\tau$ columns, each column will take $O(kn \log n)$ time, so in total it takes $O(\tau kn \log n)$ time.

Then, we know that $r(x)_{j_0} = \langle (f(x) U_b)_{j_0,*}, (U_a)_{j_0,*} \rangle$ which takes $O(\tau)$ time per $j_0$. There are $n$ different $j_0$, so it takes $O(n\tau)$ time.

Overall it takes $O(\tau kn \log n)$ time.

$\square$

**Lemma C.15** (Fat computation for $p_2(x)$). *If the following conditions hold*

- *Assume that $r(x) \in \mathbb{R}^n$ is given.*

- *Let $f(x) \in \mathbb{R}^{n \times n}$ be defined in Definition C.2.*

- *Suppose $f(x)w$ can be done in $O(kn \log n)$ time for any $w \in \mathbb{R}^n$.*

- *Let $p_2(x) = \operatorname{diag}(r(x))f(x)$ (This is obvious from definition of $r(x)$)*

*Then, we can show that*

- *For any $w \in \mathbb{R}^n$, $p_2(x) \cdot w$ can be computed $O(kn \log n)$ time.*

*Proof.* For any vector $w$, we firstly compute $f(x)w$, then we compute $\operatorname{diag}(r(x))(f(x)w)$.

$\square$

**Lemma C.16.** *If the following conditions hold*

- *Let $A_1, A_2 \in \mathbb{R}^{n \times d}$ are two given matrices.*

- *Let $p_1(x), p_2(x) \in \mathbb{R}^{n \times n}$ are defined in Definition C.7.*

- *Suppose $p_1(x)w$ takes $\mathcal{T}_{p_1}$ time for any $w \in \mathbb{R}^n$.*

- *Suppose $p_2(x)w$ takes $\mathcal{T}_{p_2}$ time for any $w \in \mathbb{R}^n$.*

*Then, we have*

- *$\operatorname{vec}(A_1^\top p(x)A_2)$ can be computed in $O(\mathcal{T}_{\mathrm{mat}}(n, d, d) + d(\mathcal{T}_{p_1} + \mathcal{T}_{p_2}))$ time.*

*Proof.* Firstly, we can compute $p_1(x)A_2$, this takes $d\mathcal{T}_{p_1}$ time.

Second, we can compute $p_2(x)A_2$, this takes $d\mathcal{T}_{p_2}$ time.

Then, we can compute $A_1^\top(p(x)A_2)$, this takes $\mathcal{T}_{\mathrm{mat}}(d, n, d) = O(\mathcal{T}_{\mathrm{mat}}(n, d, d))$.

Putting it all together we complete the proof. $\square$

### C.4 Proof of Main Theorem

In this section, we present the formal proof of our main theorem regarding the conv approximation approach for efficiently computing the training forward pass and backward gradient of the attention mechanism.

**Theorem C.17.** *Suppose $u(x)$ is a $k$-conv matrix defined in Definition 2.11 with known basis. Then there is an algorithm that runs in time $O(d^2kn \log n)$ time to compute the gradient of attention loss defined in Definition 4.1.*

*Proof.* We need to choose $\tau = d$, thus total running time is

$$\mathcal{T}_{\mathrm{mat}}(n, d, d) + O(d\tau kn \log n) = O(nd^2 k \log n),$$

by putting everything together from Lemma C.9, Lemma C.10, Lemma C.11, Lemma C.12, Lemma C.13, Lemma C.14, Lemma C.15, Lemma C.16. $\square$

**Theorem C.18** (Main conv result for training forward and backward gradient (Restatement of Theorem 4.6))**.** *If $u(x)$ is a $1/\operatorname{poly}(n)$-close $(T, \delta)$-non-degenerate $k$-conv basis matrix as defined in Definition 3.2, where $\delta \geq 0$ and $k, T \in [n]$. Then there are algorithms that run to compute **training forward** in time $O(knd \log n + \mathcal{T}_{\mathrm{mat}}(n, d, d))$ and **backward gradient** in time $O(d^2kn \log n)$ of attention loss (Definition 4.1) approximately up to $1/\operatorname{poly}(n)$ error under $\ell_\infty$ norm.*

*Proof of Theorem 4.6.* **Correctness.**

For the forward, we directly get the correctness by Theorem 3.4. For the backward, we directly run error propagation analysis which is similar to Alman & Song (2024a) and proof of Lemma B.20.

**Running time.**

For the forward, by Theorem 3.4, we directly get the running time for $D(X)^{-1}M \circ \exp(A_1 X A_2^\top)A_3$ being $O(knd \log n)$. Then, we need $\mathcal{T}_{\mathrm{mat}}(n, d, d)$ time to involve $Y$ and $E$.

For the backward, by Lemma B.20, we can use Algorithm 2 to get $k$-conv basis $\widetilde{b}_1, \ldots, \widetilde{b}_k \in \mathbb{R}^n$ and $k$ integers $m_1, m_2, \ldots, m_k$ satisfying $n \geq m_1 > m_2 > \cdots > m_k \geq T$ in time $O(knd\log(n))$. Thus, we finish the proof by Theorem C.17. □

# D INCORPORATING WEIGHTED LOW RANK APPROXIMATION

In Section D.1, we introduce the preliminary for this section. In Section D.2, we present the proof of our main result for the low-rank approximation. In Section D.3, we present the algorithm and its mathematical properties for causal attention mask. In Section D.4, we analyze the algorithm and its mathematical properties for row change by amortized constant mask. In Section D.5, we study the algorithm and its mathematical properties for continuous row mask. In Section D.6, we analyze the property of the mask matrix with $r$ distinct columns or $r$ distinct rows.

## D.1 PRELIMINARY

In this section, we introduce the background of the weighted low rank approximation.

**Definition D.1** (Definition 3.1 in Alman & Song (2023)). *Consider a positive integer $k \geq 1$. We use $\epsilon \in (0, 0.1)$ to represent an accuracy parameter. For $H \in \mathbb{R}_{\geq 0}^{n \times n}$, define $\widetilde{H} \in \mathbb{R}_{\geq 0}^{n \times n}$ to be an $(\epsilon, k)$-approximation of $H$ if*

- *$\widetilde{H}$ can be expressed as the product $U_1 \cdot U_2^\top$ with some $U_1, U_2 \in \mathbb{R}^{n \times k}$, indicating that $\widetilde{H}$ has a rank of at most $k$, and*

- *$|\widetilde{H}_{i,j} - H_{i,j}| \leq \epsilon \cdot H_{i,j}$ with any arbitrary $(i, j) \in [n] \times [n]$.*

Now, we present a lemma from Alman & Song (2023).

**Lemma D.2** (Lemma 3.4 in Alman & Song (2023)). *Let $Q, K \in \mathbb{R}^{n \times d}$ satisfy $\|Q\|_\infty \leq B$ and $\|K\|_\infty \leq B$ respectively for some $B > 0$ and $H \in \mathbb{R}^{n \times n}$ be defined as $H := \exp(QK^\top/d)$. We use $\epsilon \in (0, 0.1)$ to represent an accuracy parameter.*

*Then, there exist $g > 0$ with*

$$g = O(\max\{\frac{\log(1/\epsilon)}{\log(\log(1/\epsilon)/B^2)}, B^2\})$$

*and $k > 0$ with*

$$k \leq \binom{2(g+d)}{2g}$$

*such that: There exists an $(\epsilon, k)$-approximation (see Definition D.1) of $H \in \mathbb{R}^{n \times n}$, namely $\widetilde{H} \in \mathbb{R}^{n \times n}$. Moreover, $U_1$ and $U_2$ defining $\widetilde{H}$ is computed in $O(nk)$ time.*

In the following lemma, we prove the validity of the statement that if there exists an algorithm whose output is $Y' = (W \circ (U_1 U_2^\top))v$ in $O(t)$ time, then there exists an algorithm outputs $Y = D^{-1}(W \circ (U_1 U_2^\top))v$ in $O(t+n)$ time. We will combine everything together and show the soundness of this statement later in the proof of Theorem D.4.

**Lemma D.3.** *Let $W \in \{0, 1\}^{n \times n}$ denote any mask matrix. Let $U_1, U_2 \in \mathbb{R}^{n \times k}$. Let $v \in \mathbb{R}^n$. If there exists an algorithm whose output promises that*

$$Y' = (W \circ (U_1 U_2^\top))v,$$

*which takes $O(t)$ time, then, there exists an algorithm promise that*

$$Y = D^{-1}(W \circ (U_1 U_2^\top))v$$

*where $D := \operatorname{diag}((W \circ (U_1 U_2^\top))\mathbf{1}_n) \in \mathbb{R}^{n \times n}$, which takes $O(t+n)$ time.*

*Proof.* **Correctness.**

Suppose there exists an algorithm whose output is $Y'$ satisfying $Y' = (W \circ (U_1 U_2^\top))v$ and takes $O(t)$ time. We denote this algorithm as ALG.

Let $Y' = \text{ALG}(U_1, U_2, v)$. Let $\widetilde{Y} = \text{ALG}(U_1, U_2, \mathbf{1}_n)$. Then, $Y = \text{diag}(\widetilde{Y})^{-1} Y'$.

**Running time.**

Computing $Y'$ and $\widetilde{Y}$ takes $O(t)$ time. Computing $Y = \text{diag}(\widetilde{Y})^{-1} Y'$ takes $O(n)$ time. Therefore, it takes $O(t + n)$ time in total. $\square$

## D.2 PROOF OF MAIN RESULTS

Now, we present our main theorem.

**Theorem D.4** (Main low-rank result (Restatement of Theorem 5.5))**.** *Assume the same condition as Lemma D.2. Let $\epsilon \in (0, 0.1)$. Let $Q, K, V \in \mathbb{R}^{n \times d}$. Let $U_1, U_2 \in \mathbb{R}^{n \times k}$ be defined in Lemma D.2. Let $W \in \{0, 1\}^{n \times n}$ denote a mask matrix. Let $H = \exp(QK^\top/d) \in \mathbb{R}^{n \times n}$, $A = W \circ H \in \mathbb{R}^{n \times n}$ and $D = \text{diag}(A\mathbf{1}_n) \in \mathbb{R}^{n \times n}$. We denote $Y := D^{-1}AV \in \mathbb{R}^{n \times d}$. Let $\widetilde{A} := W \circ U_1 U_2^\top$ and $\widetilde{D} := \text{diag}(\widetilde{A}\mathbf{1}_n)$. We denote $\widetilde{Y} := \widetilde{D}^{-1}\widetilde{A}V \in \mathbb{R}^{n \times d}$. Then, we have*

$$\|Y - \widetilde{Y}\|_\infty \le 4\epsilon \|V\|_\infty.$$

*The time complexity to get $\widetilde{Y}$ is*

- *$O(knd)$ when $W$ is a causal mask defined in Definition 2.2.*

- *$O(kd \sum_{j=1}^n B_j)$ when $W$ is a row change mask defined in Definition 5.1.*

- *$O(knd \log(n))$ when $W$ is a continuous row mask defined in Definition 5.2.*

- *$O(rnd)$ when $W$ is a distinct $r$ columns / rows mask defined in Definition 5.3 / Definition 5.4.*

*Proof of Theorem 5.5.* **Correctness.**

By Lemma D.2, $U_1 U_2^\top \in \mathbb{R}^{n \times n}$ is an $(\epsilon, k)$-approximation (Definition D.1) of $H \in \mathbb{R}^{n \times n}$. Thus, we have

$$
\begin{aligned}
|\widetilde{A}_{i,j} - A_{i,j}| &= |(W \circ U_1 U_2^\top)_{i,j} - (W \circ H)_{i,j}| \\
&= W_{i,j}|(U_1 U_2^\top)_{i,j} - H_{i,j}| \\
&\le W_{i,j} \cdot \epsilon \cdot H_{i,j} \\
&= \epsilon A_{i,j},
\end{aligned}
$$

where the first step follows $\widetilde{A} = W \circ U_1 U_2^\top$ and $A = W \circ H$, the second step follows mask is element-wise operation, the third step follows Definition D.1, and the last step follows $A = W \circ H$.

Thus, by Lemma E.6, we get

$$\|Y - \widetilde{Y}\|_\infty \le 4\epsilon \|V\|_\infty.$$

**Running time.**

By Lemma D.2, the matrices $U_1$ and $U_2$ defining $\widetilde{H}$ can be computed in $O(nk)$ time.

By Lemma D.3, if we can compute $Y' = (W \circ (U_1 U_2^\top))V$ in $O(td)$ time, we can compute $\widetilde{Y}$ in $O(td + nd)$ time.

Finally, we finish the proof by following Lemma D.6 for the causal mask, Lemma D.8 for row change by amortized constant mask, Lemma D.9 for continuous row mask, and Lemma D.12 for distinct $r$ columns mask or distinct $r$ rows mask. $\square$

## D.3 CAUSAL ATTENTION MASK

In this section, we present the causal attention mask.

**Algorithm 4** Computing $(W \circ (U_1 U_2^\top))v$, where $W \in \{0,1\}^{n \times n}$ is a causal attention mask, as defined in Definition 2.2

1: **procedure** CAUSALMASK($U_1 \in \mathbb{R}^{n \times k}, U_2 \in \mathbb{R}^{n \times k}, v \in \mathbb{R}^n$)                    $\triangleright$ Lemma D.6
2:     $c_0 \leftarrow \mathbf{0}_k$
3:     **for** $j = 1 \rightarrow n$ **do**
4:         $b_j \leftarrow \underbrace{(U_2^\top)_j}_{k \times 1} \underbrace{v_j}_{\text{scalar}}$                    $\triangleright$ Let $(U_2^\top)_j$ denote the $j$-th row of $U_2 \in \mathbb{R}^{n \times k}$
5:         $c_j \leftarrow \underbrace{c_{j-1}}_{k \times 1} + \underbrace{b_j}_{k \times 1}$
6:     **end for**
7:     **for** $j = 1 \rightarrow n$ **do**
8:         $Y_j \leftarrow \langle \underbrace{(U_1^\top)_j}_{k \times 1}, \underbrace{c_j}_{k \times 1} \rangle$
9:     **end for**
10: **return** $Y$                    $\triangleright$ $Y \in \mathbb{R}^n$
11: **end procedure**

**Lemma D.5.** *Let $W \in \{0,1\}^{n \times n}$ be a mask. Let $S_j$ denote the support set of each row of $W$, for each $j \in [n]$, i.e., $S_j = \{k | W_{j,k} = 1\}$. Let $U_1, U_2 \in \mathbb{R}^{n \times k}$. Let $v \in \mathbb{R}^n$. Let $Y = (W \circ (U_1 U_2^\top))v$. Then, we have*

$$Y_j = \langle (U_1^\top)_j, \sum_{l \in S_j} (U_2^\top)_l v_l \rangle.$$

*Proof.* By simple algebra, we have

$$Y_j = ((W \circ (U_1 U_2^\top))v)_j$$
$$= \langle (U_1^\top)_j, \sum_{l \in S_j} (U_2^\top)_l v_l \rangle.$$

$\square$

**Lemma D.6.** *Let $W \in \{0,1\}^{n \times n}$ be a causal attention mask defined in Definition 2.2. Let $U_1, U_2 \in \mathbb{R}^{n \times k}$. Let $v \in \mathbb{R}^n$. Then, there exists an algorithm (see Algorithm 4) whose output promises that*

$$Y = (W \circ (U_1 U_2^\top))v,$$

*which takes $O(nk)$ time.*

*Proof.* Let $(U_2^\top)_j$ denote the $j$-th row of $U_2$.

**Correctness.**

Let $S_j$ be the support set defined in Lemma D.5. Note that for the causal attention mask, we have $S_j = [j]$ for any $j \in [n]$. Thus, by Lemma D.5, we have

$$Y_j = \langle (U_1^\top)_j, \sum_{l \in [j]} (U_2^\top)_l v_l \rangle$$
$$= \langle (U_1^\top)_j, c_j \rangle.$$

**Running time.**

Computing $(U_2^\top)_j v_j$, for all $j \in [n]$ takes $O(nk)$ time.

Note that by the definition of inner product

$$\langle (U_1^\top)_j, c_j \rangle = (U_1^\top)_j^\top c_j.$$

Therefore, it also takes $O(nk)$ to compute $(U_1^\top)_j^\top c_j$ for all $j \in [n]$.

Therefore, it takes $O(nk)$ times in total.                    $\square$

## D.4 ROW CHANGE BY AMORTIZED CONSTANT MASK

In this section, we analyze the row change by amortized constant mask.

**Claim D.7.** *Let $W \in \{0,1\}^{n \times n}$ be the causal attention mask defined in Definition 2.2. Then we have $W$ is a row change by amortized constant mask defined in Definition 5.1, where $B_j = 1, \forall j \in [n]$.*

*Proof.* The proof directly follows the two Definitions. □

---

**Algorithm 5** Computing $(W \circ (U_1 U_2^\top))v$, where $W \in \{0,1\}^{n \times n}$ is a row change by amortized constant mask, as defined in Definition 5.1

1: **procedure** CONSTANTMASK($U_1 \in \mathbb{R}^{n \times k}, U_2 \in \mathbb{R}^{n \times k}, v \in \mathbb{R}^n$)        ▷ Lemma D.8
2:      $c_0 \leftarrow \mathbf{0}_k, S_0 \leftarrow \emptyset$
3:      **for** $j = 1 \rightarrow n$ **do**
4:          Precompute indices set $Q_j^+ \leftarrow S_j \backslash S_{j-1}$ ▷ Let $S_j$ denote the support set of the $j$-th row
5:          Precompute indices set $Q_j^- \leftarrow S_{j-1} \backslash S_j$
6:          $c_j \leftarrow c_{j-1}$
7:          **for** $i \in Q_j^+ \cup Q_j^-$ **do**                    ▷ $|Q_j^+ \cup Q_j^-| = B_j$
8:              $b_i \leftarrow \underbrace{(U_2^\top)_i}_{k \times 1} \underbrace{v_i}_{\text{scalar}}$          ▷ Let $(U_2^\top)_i$ denote the $i$-th row of $U_2 \in \mathbb{R}^{n \times k}$
9:              **if** $i \in Q_j^+$ **then**
10:                  $c_j \leftarrow c_j + b_i$
11:              **else if** $i \in Q_j^-$ **then**
12:                  $c_j \leftarrow c_j - b_i$
13:              **end if**
14:          **end for**
15:      **end for**
16:      **for** $j = 1 \rightarrow n$ **do**
17:          $Y_j \leftarrow \langle \underbrace{(U_1^\top)_j}_{k \times 1}, \underbrace{c_j}_{k \times 1} \rangle$
18:      **end for**
19: **return** $Y$                            ▷ $Y \in \mathbb{R}^n$
20: **end procedure**

---

**Lemma D.8.** *Let $B \in \mathbb{Z}_{\geq 0}$ and let $W \in \{0,1\}^{n \times n}$ be a row change by amortized constant mask defined in Definition 5.1. Let $S_0 = \emptyset$. Let $S_j$ be the support set of each row of $W$, for each $j \in [n]$, i.e., $S_j = \{k | W_{j,k} = 1\}$. We define $B_j := |(S_j \backslash S_{j-1}) \cup (S_{j-1} \backslash S_j)|$. Let $U_1, U_2 \in \mathbb{R}^{n \times k}$. Let $v \in \mathbb{R}^n$. Then, there exists an algorithm (see Algorithm 5) whose output promises that*

$$Y = (W \circ (U_1 U_2^\top))v,$$

*which takes $O(k \sum_{j=1}^n B_j)$ time.*

*Proof.* **Correctness.**

By Lemma D.5, we have

$$Y_j = \langle (U_1^\top)_j, \sum_{l \in S_j} (U_2^\top)_l v_l \rangle.$$

We will prove it by induction. It is obvious that base case $Y_1$ is correct, because $S_0 = \emptyset$.

For a fixed $j$, we suppose $Y_j$ has the correct answer. This means $c_j$ is correct for that $j$, i.e., $c_j = \sum_{l \in S_j} b_l = \sum_{l \in S_j} (U_2^\top)_l v_l$.

Now we use $Q_{j+1}^+$ and $Q_{j+1}^-$ to generate $c_{j+1}$ by adding terms in $Q_{j+1}^+$ and deleting terms in $Q_{j+1}^-$,

$$c_{j+1} = \sum_{l \in S_j} b_l - \sum_{l \in S_j \backslash S_{j+1}} b_l + \sum_{l \in S_{j+1} \backslash S_j} b_l$$

$$= \sum_{l \in S_j \cap S_{j+1}} b_l + \sum_{l \in S_j \setminus S_{j+1}} b_l - \sum_{l \in S_j \setminus S_{j+1}} b_l + \sum_{l \in S_{j+1} \setminus S_j} b_l$$

$$= \sum_{l \in S_j \cap S_{j+1}} b_l + \sum_{l \in S_{j+1} \setminus S_j} b_l$$

$$= \sum_{l \in S_{j+1}} b_l,$$

where the first step follows Algorithm 5 line 10 and line 12, the second step follows $S_j = (S_j \cap S_{j+1}) \cup (S_j \setminus S_{j+1})$, $(S_j \cap S_{j+1})$ and $(S_j \setminus S_{j+1})$ are disjoint, the third step follows simple algebra, and the last step follows the as the second step.

Therefore, we have $c_{j+1}$ is correct, i.e., $c_{j+1} = \sum_{l \in S_{j+1}} b_l = \sum_{l \in S_{j+1}} (U_2^\top)_l v_l$. Thus, $Y_{j+1}$ is also correct by Lemma D.5. Finally, we finish proving the correctness by math induction.

**Running time.**

Note that there are two for-loops in this algorithm. Inside the inner for-loops, it takes $O(k)$ time to compute

$$b_i = \underbrace{(U_2^\top)_i}_{k \times 1} \underbrace{v_i}_{\text{scalar}}.$$

The inner for-loop has $|Q_j^+ \cup Q_j^-| = B_j$ iterations, and the outer for-loop has $n$ iterations.

Therefore, it takes $O(k \sum_{j=1}^n B_j)$ time in total. $\qquad\square$

### D.5 Continuous Row Mask

In this section, we study the continuous row mask.

---

**Algorithm 6** Computing $(W \circ (U_1 U_2^\top))v$, where $W \in \{0,1\}^{n \times n}$ is a continuous row mask, as defined in Definition 5.2

---

1: **procedure** CONTINUOUSMASK($U_1 \in \mathbb{R}^{n \times k}, U_2 \in \mathbb{R}^{n \times k}, v \in \mathbb{R}^n$) $\qquad\qquad \triangleright$ Lemma D.9
2: $\qquad c_0 \leftarrow \mathbf{0}_k$
3: $\qquad$ Build segment tree $\mathcal{T}$ based on $\{(U_2^\top)_i v_i\}_{i \in [n]}$
4: $\qquad$ **for** $j = 1 \rightarrow n$ **do**
5: $\qquad\qquad$ Get at most $O(\log n)$ vectors from $\mathcal{T}$ (each one is a continuous summation of $2^t$ entries)
6: $\qquad\qquad$ Compute $c_j$ based on the above vectors
7: $\qquad$ **end for**
8: $\qquad$ **for** $j = 1 \rightarrow n$ **do**
9: $\qquad\qquad Y_j \leftarrow \langle \underbrace{(U_1^\top)_j}_{k \times 1}, \underbrace{c_j}_{k \times 1} \rangle$
10: $\qquad$ **end for**
11: **return** $Y$ $\qquad\qquad\qquad\qquad\qquad\qquad\qquad\qquad\qquad\qquad \triangleright Y \in \mathbb{R}^n$
12: **end procedure**

---

**Lemma D.9.** *Let $W \in \{0,1\}^{n \times n}$ denote a continuous row mask defined in Definition 5.2. Let $U_1, U_2 \in \mathbb{R}^{n \times k}$. Let $v \in \mathbb{R}^n$. Then, there exists an algorithm (see Algorithm 6) whose output promises that*

$$Y = (W \circ (U_1 U_2^\top))v,$$

*which takes $O(nk \log n)$ time.*

*Proof.* The correctness is trivially from the construction of the segment tree.

The running time is dominated by $O(nk \log n)$. This time comes from two parts, where the first is from building the segment tree by $O(nk)$, and the second part is from for-loop by $O(nk \log n)$. $\quad\square$

### D.6 DISTINCT $r$ COLUMNS OR ROWS

Now, we analyze the mask matrix with $r$ distinct columns.

**Lemma D.10.** *Let $W$ be the distinct $r$ columns mask defined in Definition 5.3. Let $S_1, \cdots, S_r \subseteq [n]$ denote $r$ disjoint subsets and $\cup_{j \in [r]} S_j = [n]$ be defined in Definition 5.3. Let $h : [r] \to [n]$ denote that $h(j) \in S_j$ and $h(j)$ is the smallest index in $S_j$.*

*Then we can show*

$$\underbrace{(W}_{n \times n} \circ (\underbrace{U_1}_{n \times k} \underbrace{U_2^\top}_{k \times n})) \underbrace{v}_{n \times 1} = \sum_{j=1}^r \underbrace{\mathrm{diag}(W_{*,h(j)})}_{n \times n} \underbrace{U_1}_{n \times k} \underbrace{(U_2^\top)_{*,S_j}}_{k \times |S_j|} \underbrace{v_{S_j}}_{|S_j| \times 1}$$

*Proof.* We can show that

$$\begin{aligned}
\mathrm{LHS} &= \sum_{i=1}^n (W \circ (U_1 U_2^\top))_{*,i} \cdot v_i \\
&= \sum_{i=1}^n (W_{*,i} \circ (U_1 U_2^\top)_{*,i}) v_i \\
&= \sum_{i=1}^n \mathrm{diag}(W_{*,i})(U_1 U_2^\top)_{*,i} v_i \\
&= \sum_{i=1}^n \mathrm{diag}(W_{*,i}) U_1 (U_2^\top)_{*,i} v_i \\
&= \sum_{j=1}^r \mathrm{diag}(W_{*,h(j)}) U_1 (U_2^\top)_{*,S_j} v_{S_j},
\end{aligned}$$

where the first step follows from the left hand side of the equation in the lemma statement, the second step follows from the definition of the Hadamard product, the third step follows from Fact B.5, the fourth step follows from simple algebra, and the last step follows from the fact that for any two $i, i' \in S_j$, we have $W_{*,i} = W_{*,i'} \in \mathbb{R}^n$ (see from the lemma statement). □

Now, we analyze the mask matrix with $r$ distinct rows.

**Lemma D.11.** *Let $W$ be the distinct $r$ rows mask defined in Definition 5.4. Let $S_1, \cdots, S_r \subseteq [n]$ denote $r$ disjoint subsets and $\cup_{j \in [r]} S_j = [n]$ be defined in Definition 5.4. Let $h : [r] \to [n]$ denote that $h(j) \in S_j$ and $h(j)$ is the smallest index in $S_j$.*

*Then, we can show that*

$$\underbrace{(W}_{n \times n} \circ (\underbrace{U_1}_{n \times k} \underbrace{U_2^\top}_{k \times n})) \underbrace{v}_{n \times 1} = \sum_{j=1}^r \underbrace{\mathrm{diag}(e_{S_j})}_{n \times n} \underbrace{U_1}_{n \times k} \underbrace{U_2^\top}_{k \times n} \underbrace{\mathrm{diag}(W_{h(j),*})}_{n \times n} \underbrace{v}_{n \times 1}$$

*Proof.* It suffices to show

$$\underbrace{(W}_{n \times n} \circ (\underbrace{U_1}_{n \times k} \underbrace{U_2^\top}_{k \times n})) = \sum_{j=1}^r \underbrace{\mathrm{diag}(e_{S_j})}_{n \times n} \underbrace{U_1}_{n \times k} \underbrace{U_2^\top}_{k \times n} \underbrace{\mathrm{diag}(W_{h(j),*})}_{n \times n}. \tag{10}$$

We have

$$\begin{aligned}
(W \circ (U_1 U_2^\top)) &= ((U_1 U_2^\top) \circ W) \\
&= \sum_{i=1}^n (\mathrm{diag}(e_i)(U_1 U_2^\top) \circ W)_{i,*} \\
&= \sum_{i=1}^n (\mathrm{diag}(e_i)(U_1 U_2^\top) \circ W_{i,*})
\end{aligned}$$

$$= \sum_{i=1}^{n} (\mathrm{diag}(e_i)(U_1 U_2^\top) \, \mathrm{diag}(W_{i,*}))$$

$$= \sum_{j=1}^{n} \mathrm{diag}(e_{S_j}) U_1 U_2^\top \, \mathrm{diag}(W_{h(j),*}),$$

where the first step follows from the definition of the Hadamard product, the second step follows from the property of $\mathrm{diag}(e_i)$ that for any matrix $A$, $\mathrm{diag}(e_i)A$ preserves the $i$-th row of $A$ and set other rows to 0, the third step follows from simple algebra, the fourth step follows from Fact B.5, and the last step follows from the lemma statement that for any two $i, i' \in S_j$, we have $W_{i,*} = W_{i',*} \in \mathbb{R}^n$.

Therefore, we have shown Eq. (10), which completes the proof. $\qquad\square$

**Lemma D.12.** *Let $W \in \{0,1\}^{n \times n}$ be a distinct $r$ columns mask defined in Definition 5.3 or a distinct $r$ rows mask defined in Definition 5.4. Let $U_1, U_2 \in \mathbb{R}^{n \times k}$. Let $v \in \mathbb{R}^n$. Then, there exists an algorithm whose output promises that*

$$Y = (W \circ (U_1 U_2^\top))v,$$

*which takes $O(nkr)$ time.*

*Proof.* The correctness and running time is directly follows Lemma D.10 for the column case and Lemma D.11 for the row case. $\qquad\square$

# E SUPPORTING LEMMAS AND TECHNICAL RESULTS

In Section E.1, we present the matrix and vector properties. In Section E.2, we analyze and develop the tools for error analysis. In Section E.3, we provide some tools for tensor calculation.

## E.1 MATRIX AND VECTOR PROPERTIES

**Lemma E.1** (Restatement of Lemma 2.12). *For any lower triangular matrix $H \neq \mathbf{0}_{n \times n} \in \mathbb{R}^{n \times n}$, there exists a unique $k \in [n]$ such that $H$ is a matrix with $k$-conv basis.*

*Proof of Lemma 2.12.* It suffices to show that any arbitrary $H \in \mathbb{R}^{n \times n} \setminus \{\mathbf{0}_{n \times n}\}$ has at least 1 conv basis and at most $n$ conv basis.

As $H \neq \mathbf{0}_{n \times n}$, it must have at least 1 conv basis, and we proved the first part.

Now, we prove the second part by math induction.

Let $i \in \{0, \ldots, n-1\}$. For any lower triangular matrix $G \in \mathbb{R}^{n \times n}$, we have

$$G = \begin{bmatrix} \mathbf{0}_{i \times i} & \mathbf{0}_{i \times (n-i)} \\ \mathbf{0}_{(n-i) \times i} & G_{(i+1):n,(i+1):n} \end{bmatrix}.$$

Let $G_{i+1}$ be the $i+1$-th column of $G \in \mathbb{R}^{n \times n}$. Let $\widetilde{G}_{i+1} \in \mathbb{R}^n$ satisfy, for any $j \in [n]$, $(\widetilde{G}_{i+1})_j = (G_{i+1})_{i+j}$ when $i + j \leq n$ and $(\widetilde{G}_{i+1})_j = (G_{i+1})_{i+j-n}$ otherwise. Then, there exists lower triangular matrix $G' \in \mathbb{R}^{(n-i-1) \times (n-i-1)}$ such that

$$G - \mathsf{conv}(\widetilde{G}_{i+1}, n - i)$$

$$= \begin{bmatrix} \mathbf{0}_{i \times i} & \mathbf{0}_{i \times 1} & \mathbf{0}_{i \times (n-i-1)} \\ \mathbf{0}_{1 \times i} & G_{i+1,i+1} & \mathbf{0}_{1 \times (n-i-1)} \\ \mathbf{0}_{(n-i-1) \times i} & G_{(i+2):n,(i+1)} & G_{(i+2):n,(i+2):n} \end{bmatrix} - \begin{bmatrix} \mathbf{0}_{i \times i} & \mathbf{0}_{i \times 1} & \mathbf{0}_{i \times (n-i-1)} \\ \mathbf{0}_{1 \times i} & G_{i+1,i+1} & \mathbf{0}_{1 \times (n-i-1)} \\ \mathbf{0}_{(n-i-1) \times i} & G_{(i+2):n,(i+1)} & G' \end{bmatrix}$$

$$= \begin{bmatrix} \mathbf{0}_{i \times i} & \mathbf{0}_{i \times 1} & \mathbf{0}_{i \times (n-i-1)} \\ \mathbf{0}_{1 \times i} & \mathbf{0}_{1 \times 1} & \mathbf{0}_{1 \times (n-i-1)} \\ \mathbf{0}_{(n-i-1) \times i} & \mathbf{0}_{(n-i-1) \times 1} & G_{(i+2):n,(i+2):n} - G' \end{bmatrix}$$

$$= \begin{bmatrix} \mathbf{0}_{(i+1) \times (i+1)} & \mathbf{0}_{(i+1) \times (n-i-1)} \\ \mathbf{0}_{(n-i-1) \times (i+1)} & G_{(i+2):n,(i+2):n} - G' \end{bmatrix},$$

where the first step follows from the fact that $G$ is a lower triangular matrix and Definition 2.9, the second step follows from simple algebra, and the last step follows from simple algebra.

As $G$ and $G'$ are lower triangular matrices, we have that $G - \mathsf{conv}(\widetilde{G}_{i+1}, n - i)$ is a lower triangular matrix. Thus, we proved the following statement.

For any lower triangular matrix $G \in \mathbb{R}^{n \times n}$ whose first $i$ columns all are zeros, there exists a basis $\mathsf{conv}(b, m)$ such that $G - \mathsf{conv}(b, m) \in \mathbb{R}^{n \times n}$ is a lower triangular matrix whose first $i + 1$ columns all are zeros.

As $H \in \mathbb{R}^{n \times n}$ is a lower triangular matrix whose first $0$ columns all are zeros, we finish the proof by math induction, i.e., repeat the above process at most $n$ times. $\qquad\square$

**Lemma E.2.** *For any matrix $G \in \mathbb{R}^{n \times n}$ and vector $v \in \mathbb{R}^n$, we have*

$$\|Gv\|_1 \le \|G\|_1 \cdot \|v\|_\infty.$$

*Proof.* We have

$$\begin{aligned}
\|Gv\|_1 &= \sum_{i \in [n]} |\sum_{j \in [n]} G_{i,j} v_j| \\
&\le \sum_{i \in [n]} \sum_{j \in [n]} |G_{i,j} v_j| \\
&\le \sum_{i \in [n]} \sum_{j \in [n]} |G_{i,j}| \|v\|_\infty \\
&= \|G\|_1 \cdot \|v\|_\infty,
\end{aligned}$$

where the first step follows the Definition of vector $\ell_1$ norm, the second steps follow $|a + b| \le |a| + |b|$, the third steps follow simple algebra, and the last step follow the Definition of matrix $\ell_1$ norm. $\quad\square$

### E.2 Tools for Error Analysis

**Lemma E.3.** *Let $\epsilon \ge 0$. Let $x_1, x_2 \in \mathbb{R}$. We have*

$$|\exp(x_1) - \exp(x_2)| \le \exp(\min\{x_1, x_2\})(\exp(|x_1 - x_2|) - 1).$$

*Proof.* It is trivial by $\exp(a + b) = \exp(a) \exp(b)$. $\qquad\square$

**Lemma E.4.** *Let $V \in \mathbb{R}^{n \times d}$. Let $H, \widetilde{H} \in \mathbb{R}^{n \times n}$, and satisfy $\|H - \widetilde{H}\|_\infty \le \epsilon$, where $\epsilon \ge 0$. Let $A = \exp(H)$, $\widetilde{A} = \exp(\widetilde{H})$ and $D = \mathrm{diag}(A\mathbf{1}_n)$, $\widetilde{D} = \mathrm{diag}(\widetilde{A}\mathbf{1}_n)$. Then, we have*

$$\|D^{-1}AV - \widetilde{D}^{-1}\widetilde{A}V\|_\infty \le 2(\exp(\epsilon) - 1)\|V\|_\infty.$$

*Proof.* By triangle inequality, we have

$$\|D^{-1}AV - \widetilde{D}^{-1}\widetilde{A}V\|_\infty = \|D^{-1}AV - \widetilde{D}^{-1}AV\|_\infty + \|\widetilde{D}^{-1}AV - \widetilde{D}^{-1}\widetilde{A}V\|_\infty,$$

where the first step follows simple algebra, and the last step follows triangle inequality.

For the first part, for any $i \in [n], j \in [n]$, we have

$$\begin{aligned}
|(D^{-1}AV - \widetilde{D}^{-1}AV)_{i,j}| &= |\sum_{l=1}^n (D_{i,i}^{-1} - \widetilde{D}_{i,i}^{-1}) A_{i,l} V_{l,j}| \\
&\le \sum_{l=1}^n |(D_{i,i}^{-1} - \widetilde{D}_{i,i}^{-1}) A_{i,l}| \cdot \|V\|_\infty \\
&= \sum_{l=1}^n |\frac{D_{i,i} - \widetilde{D}_{i,i}}{D_{i,i} \widetilde{D}_{i,i}}| \cdot A_{i,l} \cdot \|V\|_\infty
\end{aligned}$$

$$= \sum_{l=1}^{n} |\sum_{k=1}^{n} \exp(H_{i,k}) - \sum_{k=1}^{n} \exp(\widetilde{H}_{i,k})| \cdot \frac{A_{i,l}}{D_{i,i}\widetilde{D}_{i,i}} \cdot \|V\|_{\infty}$$

$$\leq \sum_{l=1}^{n} \sum_{k=1}^{n} |\exp(H_{i,k}) - \exp(\widetilde{H}_{i,k})| \cdot \frac{A_{i,l}}{D_{i,i}\widetilde{D}_{i,i}} \cdot \|V\|_{\infty}$$

$$\leq (\exp(\epsilon) - 1) \sum_{l=1}^{n} \sum_{k=1}^{n} \exp(\widetilde{H}_{i,k}) \cdot \frac{A_{i,l}}{D_{i,i}\widetilde{D}_{i,i}} \cdot \|V\|_{\infty}$$

$$= (\exp(\epsilon) - 1)\|V\|_{\infty},$$

where the first step follows simple algebra, the second step follows triangle inequality, the third step follows simple algebra, the fourth step follows $D = \mathrm{diag}(A\mathbf{1}_n)$, $\widetilde{D} = \mathrm{diag}(\widetilde{A}\mathbf{1}_n)$, $A = \exp(H)$, $\widetilde{A} = \exp(\widetilde{H})$, the fifth steps follows triangle inequality, the sixth step follows Lemma E.3 and the last step follows $\widetilde{D}_{i,i} = \sum_{k=1}^{n} \exp(\widetilde{H}_{i,k})$ and $D_{i,i} = \sum_{l=1}^{n} A_{i,l}$.

For the second part, for any $i \in [n], j \in [n]$, we have

$$|(\widetilde{D}^{-1}AV - \widetilde{D}^{-1}\widetilde{A}V)_{i,j}| = |\sum_{l=1}^{n} \widetilde{D}_{i,i}^{-1}(A_{i,l} - \widetilde{A}_{i,l})V_{l,j}|$$

$$\leq \sum_{l=1}^{n} \widetilde{D}_{i,i}^{-1}|A_{i,l} - \widetilde{A}_{i,l}| \cdot \|V\|_{\infty}$$

$$= \sum_{l=1}^{n} \widetilde{D}_{i,i}^{-1}|\exp(H_{i,l}) - \exp(\widetilde{H}_{i,l})| \cdot \|V\|_{\infty}$$

$$\leq (\exp(\epsilon) - 1) \sum_{l=1}^{n} \widetilde{D}_{i,i}^{-1} \exp(\widetilde{H}_{i,l}) \cdot \|V\|_{\infty}$$

$$= (\exp(\epsilon) - 1)\|V\|_{\infty},$$

where the first step follows simple algebra, the second step follows triangle inequality, the third step follows $A = \exp(H)$, $\widetilde{A} = \exp(\widetilde{H})$, the fourth step follows Lemma E.3, and the last step follows $\widetilde{D}_{i,i} = \sum_{l=1}^{n} \exp(\widetilde{H}_{i,l})$.

Thus, we combine two terms,

$$\|D^{-1}AV - \widetilde{D}^{-1}\widetilde{A}V\|_{\infty} \leq 2(\exp(\epsilon) - 1)\|V\|_{\infty}.$$

$\square$

**Lemma E.5.** *Let $a, b \geq 0$ and $\epsilon \in (0, 0.1)$. If $|a - b| \leq \epsilon a$, then $|a - b| \leq 2\epsilon \min\{a, b\}$.*

*Proof.* It is trivial by considering two cases when $b \geq a$ and $b < a$. $\square$

**Lemma E.6.** *Let $A, \widetilde{A} \in \mathbb{R}_{\geq 0}^{n \times n}$, and satisfy $|\widetilde{A}_{i,j} - A_{i,j}| \leq \epsilon \cdot A_{i,j}$ for all $(i, j) \in [n]^2$, where $\epsilon \in (0, 0.1)$. Let $D = \mathrm{diag}(A\mathbf{1}_n)$ and $\widetilde{D} = \mathrm{diag}(\widetilde{A}\mathbf{1}_n)$. Then, we have*

$$\|D^{-1}AV - \widetilde{D}^{-1}\widetilde{A}V\|_{\infty} \leq 4\epsilon\|V\|_{\infty}.$$

*Proof.* By triangle inequality, we have

$$\|D^{-1}AV - \widetilde{D}^{-1}\widetilde{A}V\|_{\infty} \leq \|D^{-1}AV - \widetilde{D}^{-1}AV\|_{\infty} + \|\widetilde{D}^{-1}AV - \widetilde{D}^{-1}\widetilde{A}V\|_{\infty},$$

where the first step follows simple algebra, and the last step follows triangle inequality.

For the first part, for any $i \in [n], j \in [n]$, we have

$$|(D^{-1}AV - \widetilde{D}^{-1}AV)_{i,j}| = |\sum_{l=1}^{n} (D_{i,i}^{-1} - \widetilde{D}_{i,i}^{-1})A_{i,l}V_{l,j}|$$

$$\leq \sum_{l=1}^{n} |(D_{i,i}^{-1} - \widetilde{D}_{i,i}^{-1}) A_{i,l}| \cdot \|V\|_\infty$$

$$= \sum_{l=1}^{n} |\frac{D_{i,i} - \widetilde{D}_{i,i}}{D_{i,i} \widetilde{D}_{i,i}}| \cdot A_{i,l} \cdot \|V\|_\infty$$

$$= \sum_{l=1}^{n} |\sum_{k=1}^{n} A_{i,k} - \sum_{k=1}^{n} \widetilde{A}_{i,k}| \cdot \frac{A_{i,l}}{D_{i,i} \widetilde{D}_{i,i}} \cdot \|V\|_\infty$$

$$\leq \sum_{l=1}^{n} \sum_{k=1}^{n} |A_{i,k} - \widetilde{A}_{i,k}| \cdot \frac{A_{i,l}}{D_{i,i} \widetilde{D}_{i,i}} \cdot \|V\|_\infty$$

$$\leq 2\epsilon \sum_{l=1}^{n} \sum_{k=1}^{n} \widetilde{A}_{i,k} \cdot \frac{A_{i,l}}{D_{i,i} \widetilde{D}_{i,i}} \cdot \|V\|_\infty$$

$$= 2\epsilon \|V\|_\infty,$$

where the first step follows simple algebra, the second step follows triangle inequality, the third step follows simple algebra, the fourth step follows $D = \mathrm{diag}(A\mathbf{1}_n)$, $\widetilde{D} = \mathrm{diag}(\widetilde{A}\mathbf{1}_n)$, the fifth step follows triangle inequality, the sixth step follows Lemma E.5 and the last step follows $\widetilde{D}_{i,i} = \sum_{k=1}^{n} \widetilde{A}_{i,k}$ and $D_{i,i} = \sum_{l=1}^{n} A_{i,l}$.

For the second part, for any $i \in [n], j \in [n]$, we have

$$|(\widetilde{D}^{-1} AV - \widetilde{D}^{-1} \widetilde{A} V)_{i,j}| = |\sum_{l=1}^{n} \widetilde{D}_{i,i}^{-1} (A_{i,l} - \widetilde{A}_{i,l}) V_{l,j}|$$

$$\leq \sum_{l=1}^{n} \widetilde{D}_{i,i}^{-1} |A_{i,l} - \widetilde{A}_{i,l}| \cdot \|V\|_\infty$$

$$\leq 2\epsilon \sum_{l=1}^{n} \widetilde{D}_{i,i}^{-1} \widetilde{A}_{i,l} \cdot \|V\|_\infty$$

$$= 2\epsilon \|V\|_\infty,$$

where the first step follows simple algebra, the second step follows triangle inequality, the third step follows Lemma E.5, and the last step follows $\widetilde{D}_{i,i} = \sum_{l=1}^{n} \widetilde{A}_{i,l}$.

Thus, we combine two terms,

$$\|D^{-1} AV - \widetilde{D}^{-1} \widetilde{A} V\|_\infty \leq 4\epsilon \|V\|_\infty.$$

$\square$

### E.3 TENSOR TOOLS FOR GRADIENT COMPUTATION

**Fact E.7** (Fact A.3 on page 15 of Li et al. (2024c), also see Bürgisser et al. (2013); Bläser (2013) for more detail). *We can show that*

$$\mathcal{T}_{\mathrm{mat}}(a,b,c) = O(\mathcal{T}_{\mathrm{mat}}(a,c,b)) = O(\mathcal{T}_{\mathrm{mat}}(b,a,c)) = O(\mathcal{T}_{\mathrm{mat}}(b,c,a)) = O(\mathcal{T}_{\mathrm{mat}}(c,a,b)) = O(\mathcal{T}_{\mathrm{mat}}(c,b,a)).$$

**Fact E.8.** *Let $a \in \mathbb{R}^n, b \in \mathbb{R}^d$. We have*

$$\mathrm{vec}(ab^\top) = a \otimes b$$

*Proof.* We can show

$$\mathrm{vec}(ab^\top) = \mathrm{vec}(\begin{bmatrix} a_1 b^\top \\ a_2 b^\top \\ \dots \\ a_n b^\top \end{bmatrix})$$

$$= [a_1 b^\top, a_2 b^\top, \dots, a_n b^\top]^\top$$

$$= a \otimes b$$

where the first step follows from the definition of outer product, the second step follows from the definition of vectorization operator $\mathrm{vec}(\cdot)$ which stacks rows of a matrix into a column vector, and the last step follows from Definition 4.4. □

**Fact E.9** (Tensor-trick on page 3 of Gao et al. (2023a), also see Diao et al. (2018) for more detail)**.** *Given matrices* $A_1 \in \mathbb{R}^{n_1 \times d_1}$, $A_2 \in \mathbb{R}^{n_2 \times d_2}$ *and* $X \in \mathbb{R}^{d_1 \times d_2}$, *the well-known tensor-trick suggests that* $\mathrm{vec}(A_1 X A_2^\top) = (A_1 \otimes A_2) \mathrm{vec}(X) \in \mathbb{R}^{n_1 n_2}$.

*Proof.* We can show

$$\mathrm{vec}(A_1 X A_2^\top) = \sum_{i=1}^{d_1} \sum_{j=1}^{d_2} X_{i,j} \mathrm{vec}(A_{1,*,i}(A_{2,*,j})^\top)$$

$$= \sum_{i=1}^{d_1} \sum_{j=1}^{d_2} X_{i,j} (\underbrace{A_{1,*,i}}_{n_1 \times 1} \otimes \underbrace{A_{2,*,j}}_{n_2 \times 1})$$

$$= \sum_{i=1}^{d_1} (\underbrace{A_{1,*,i}}_{n_1 \times 1} \otimes \underbrace{A_2}_{n_2 \times d_2}) \underbrace{X_{i,*}}_{d_2 \times 1}$$

$$= (A_1 \otimes A_2) \mathrm{vec}(X)$$

where the first step follows from that matrix can be written as a summation of vectors, the second step follows from Fact E.8, the third step follows from that matrix can be written as a summation of vectors, and the last step follows from the definition of vectorization operator $\mathrm{vec}(\cdot)$. □

## F    MORE RELATED WORK

**Fast attention computation and long context LLM.**    The development of efficient attention computation has been an active area of research in recent years. The standard self-attention mechanism, introduced in the transformer architecture (Vaswani et al., 2017), has a quadratic complexity with respect to the sequence length, which limits its applicability to long sequences. To address this limitation, various approaches have been proposed to improve the efficiency of attention computation. One line of research focuses on patterns of sparse attention that reduce the number of computations (Child et al., 2019; Beltagy et al., 2020; Zaheer et al., 2020; Shi et al., 2023a; Han et al., 2024). Another approach is to use low-rank approximations or random features for the attention matrix (Razenshteyn et al., 2016; Li et al., 2016; Wang et al., 2020; Choromanski et al., 2020; Zheng et al., 2022; Alman & Song, 2023; Ahn et al., 2024), which reduces the computational complexity to linear in the sequence length. In addition, using linear attention as a proxy of Softmax attention is a rich line of work (Tsai et al., 2019; Katharopoulos et al., 2020; Schlag et al., 2021; Zhang et al., 2023; Sun et al., 2023; Ahn et al., 2024; Shi et al., 2023b; Xu et al., 2024b; Zhang et al., 2024; Deng et al., 2023). These developments in efficient attention computation have enabled transformer-based models to process longer sequences and have opened up new possibilities for their application in various domains (Chen et al., 2023b; Su et al., 2024; Peng et al., 2024; Ding et al., 2024; Ma et al., 2024; Xu et al., 2024c; An et al., 2024; Bertsch et al., 2024; Chen et al., 2024; Liang et al., 2024d; Jin et al., 2024; Shi et al., 2024).

**Convolution in language model and FFT.**    There are many subquadratic-time architectures are proposed to address Transformers' computational inefficiency on long sequences, gated convolution recurrent models (Bai et al., 2018; Fu et al., 2023; Peng et al., 2023; Qin et al., 2023), and structured state space models (SSMs) (Gu et al., 2021; Gu & Dao, 2023). They can use global or local convolution (Krizhevsky et al., 2012) operations to replace attention while keeping a comparable performance. The convolution operation can be computed by fast Fourier transform (FFT) efficiently (Pratt et al., 2017; Chi et al., 2020). Moreover, the development of efficient convolution algorithms like Winograd (Lavin & Gray, 2016) and FFT-based convolutions (Mathieu et al., 2013) has further optimized the computation, reducing the memory footprint and improving the overall speed. There are many other works studying Fourier transform (Price & Song, 2015; Moitra, 2015; Chen et al., 2016; Song, 2019;

Lee et al., 2019; Chen et al., 2020; Song et al., 2022; Gao et al., 2022; Song et al., 2023a; Chen et al., 2023a; Song et al., 2023d; Jin et al., 2023).

**(Weighted) low rank approximation.** Low-rank approximation has become an important tool in machine learning and numerical linear algebra, providing a way to extract the core structure of high-dimensional data while minimizing computational costs. Mathematically, we want to find matrices $X, Y \in \mathbb{R}^{n \times k}$ such that $\|M - XY^\top\|_F$ is minimized. It has been applied to various fields, such as training multi-layer neural network Song et al. (2021), attention approximation Alman & Song (2023; 2024a), dynamic Kronecker product maintenance Song et al. (2023c), and tensor product regression Reddy et al. (2022). In practice, certain entries of $M$ tend to be more important than others, leading to the study of the weighted low-rank approximation: finding matrices $X, Y \in \mathbb{R}^{n \times k}$ such that $\|W \circ (M - XY^\top)\|_F$ is minimized, where $W \in \mathbb{R}_{\geq 0}^{n \times n}$ Li et al. (2016); Razenshteyn et al. (2016); Song et al. (2023e); Gu et al. (2024). As data continues to grow in size and complexity, (weighted) low rank approximation remains an active area of research, with ongoing efforts to develop more efficient, scalable, and robust methods for a wide range of applications.

**Attention optimization.** There are several other techniques optimizing the approximation of the attention computation to alleviate the quadratic complexity $O(n^2)$, such as optimizing the attention-related regression problems Song et al. (2023f); Gao et al. (2023b;c;d); Li et al. (2024b); Liang et al. (2024b), multi-layer attention optimization Song et al. (2023b); Li et al. (2023b); Liang et al. (2024c), cross attention Liang et al. (2024f), Hopfield Models (Hu et al., 2023; Wu et al., 2024b; Hu et al., 2024c; Xu et al., 2024a; Wu et al., 2024a; Hu et al., 2024a;b;d), and optimizing the tensor version of the attention approximation Liang et al. (2024e); Alman & Song (2024b).

