# OpenReview forum: "Conv-Basis: A New Paradigm for Efficient Attention Inference and Gradient Computation in Transformers"
_ICLR.cc/2025/Conference — Submitted to ICLR 2025_

### Official Review · Reviewer_gXgk · 2024-11-04

**Soundness:** 2
**Presentation:** 2
**Contribution:** 2
**Rating:** 3
**Confidence:** 3

**Summary:**

This work investigates the problem of approximating matrices using a lower-triangular Toeplitz matrix basis, or "conv-basis", focusing specifically on the self-attention mechanism. The authors provide theoretical results about the approximability of lower triangular matrices into such a convolution basis, and provide algorithms for efficiently recovering this basis and using it to efficiently perform the self-attention operation using FFTs. Finally, the authors provide empirical results showing that their proposed method requires few basis elements to recover performance on the IMDB dataset when applied to Llama3 8B.

**Strengths:**

- The authors investigate an important problem: how to speed up the efficiency of the self-attention mechanism, the computational bottleneck in Transformers.
- The theoretical results are interesting, thorough, and clearly presented.

**Weaknesses:**

- I think the theoretical results are quite interesting and deserve to be presented. However, the main weakness seems to be that the empirical results are quite limited.
  - The empirical speedup results are limited. Although asymptotically, the authors argue that the conv-basis approximation is more efficient, as far as I can tell, an empirical comparison of their method's speed vs. standard self-attention is not provided. It could be clarifying to provide such a comparison in the case of e.g. Llama3 8B, so that it's clear at what sequence length such an approach becomes more efficient in practice.
  - Additionally, the expressivity vs. efficiency tradeoff inherent to the method could be explored more thoroughly. The main theoretical result in the paper is that any lower triangular matrix can be well approximated in the conv basis, but accurate recovery may take as many as n basis elements in the worst case. The authors further support this result with experimental validation on the IMDB dataset with Llama3 8B, but this is only one task and relatively simple. It could be clarifying to provide additional experimental results on more recent LM benchmarks (e.g. HellaSwag, WinoGrande?)

**Questions:**

- As the authors mention, the proposed concept of the conv-basis approximation to sequence models seems quite related to prior work around sub-quadratic alternatives to attention, e.g. SSMs [1], Mamba [2]. Is there some way a conv-approximation of an existing attention matrix can be interpreted as a set of specific kernels for a subquadratic model?
  - In particular, I wonder if the conv-basis concept relates to the semiseparable matrices of Mamba2 [3]?
- It would be interesting to see how the approximability of attention vs. number of conv-basis elements changes based on the task, e.g. do math reasoning tasks (GSM8K) require more basis elements than simple question-answering?

1. Efficiently Modeling Long Sequences with Structured State Spaces
2. Mamba: Linear-Time Sequence Modeling with Selective State Spaces
3. Transformers are SSMs: Generalized Models and Efficient Algorithms Through Structured State Space Duality

---

> ### Author Response · Authors · 2024-11-26
>
> We express our deepest gratitude to the reviewer for the time and effort in reviewing our work. It is very encouraging to see that our theoretical results are regarded as comprehensive, interesting, and well-presented. Additionally, the problem we study is recognized as important. Below, we address your concerns regarding our work.
>
> ### W1 & Q2: Empirical results
> Thank you for your question. We would like to emphasize that our work is theoretical in nature. We have provided a detailed comparison of our work with recent papers, such as [1] and [2], in Line 129–145. Our motivation stems from the fact that their assumptions are too restrictive, making their algorithms challenging to implement successfully. In contrast, our work develops an implementable, assumption-light algorithm that can be applied in real-world scenarios while maintaining a comparable running time to [1] and [2]. The experiment in our paper serves as an example to demonstrate the implementation of our theoretical contribution and to highlight its independence from the restrictive assumptions of [1]. However, it is not the primary focus of our work. Our experiment illustrates that our theoretical results can be directly implemented in a popular transformer model, Llama 3, whereas [1] lacks any experimental validation. Additionally, our work is theoretically comparable to—and, in many aspects, surpasses—the prior theoretical contributions of [1] and [2] (see Line 129–145 and our response to Reviewer vPbo for further details). We also refer the reviewer to the Global response **Empirical evaluation** section for more details.
>
> ### Q1: Is there some way a conv-approximation of an existing attention matrix can be interpreted as a set of specific kernels for a subquadratic model? In particular, I wonder if the conv-basis concept relates to the semiseparable matrices of Mamba2 [3]?
>
> Thank you for your brilliant question. The conv basis $H = \sum_{i} \mathrm{conv}(b_i, m_i)$ can be viewed as learning $m_i$ length-selective filters $b_i$ that capture different ranges of dependencies in the sequence. This connects to how SSMs use structured state transitions to model dependencies efficiently. While related, our approach differs from Mamba's semiseparable matrices.
> - When defining semiseparable matrices, they rely on the matrix rank. However, our convolution matrix definition is completely unrelated to rank. We are more related to the Toeplitz matrix, in Figure 3 of Mamba2 [3].
> - Mamba2 can be viewed as linear attention in some sense, while our work is approximating standard attention.
> - The theoretical guarantees on approximation error are different.
>
> We thank you again for your insightful comments.
>
> [1] Josh Alman, and Zhao Song. "Fast attention requires bounded entries." NeurIPS’23.
>
> [2] Insu Han, Rajesh Jayaram, Amin Karbasi, Vahab Mirrokni, David P. Woodruff, and Amir Zandieh. "Hyperattention: Long-context attention in near-linear time." ICLR’24.
>
> [3] Dao, Tri, and Albert Gu. "Transformers are SSMs: Generalized models and efficient algorithms through structured state space duality." ICML’24.

---

> > ### Comment · Reviewer_gXgk · 2024-11-30
> >
> > Thanks to the authors for the detailed response and revisions! I appreciate the emphasis that this work is theoretical in nature, and that the theoretical results improve upon [1] and [2] and thus may be interesting to the community. I would be willing to increase my score if one of the other reviewers strongly advocates for acceptance based on the strength of the theoretical results.
> >
> > I'm still reluctant to advocate for acceptance due to the limited empirical results. I understand that the work is mainly theoretical in nature, but I believe additional empirical results would strengthen this work significantly:
> > 1. The framing of the work repeatedly emphasizes that the theoretical findings are motivated towards speeding up attention in practice. For example, "our paradigm for accelerating attention computation" (line 27), the motivating question in lines 89-90, the claim that "our algorithm can accelerate Transformer training as well. It may save time, resources, and energy for nowadays LLMs training" in line 431, and even the title of the work about "efficient attention inference and gradient computations". As such, it seems like a comparison of runtimes, or at least of FLOPs, vs. standard attention implementations in practice would be crucial to validate claims about efficiency or accelerating attention. Without such results, some of these claims in the paper seem overstated.
> > 2. I understand that a properly efficient implementation of the conv-basis algorithms would be time-consuming and out-of-scope for the paper. However, as Reviewer Z8SA also points out, it seems to me that there are several basic experimental verifications that are crucial to the argument of the paper, which are not included:
> > - The main result in Section 4 about training proves efficient training algorithms for attention, but there are no experimental results using the conv-basis during training. A verification that the conv-basis paradigm is effective for training, and especially a comparison of runtime/FLOPs of standard attention vs. conv-basis backwards gradient computation would strengthen the paper significantly.
> > - Furthermore, we know that the conv-basis paradigm should see significant speedups relative to naive attention as sequence length increases, but it's unclear where that tradeoff currently occurs. A comparison of runtimes or FLOPs with a standard attention implementation as sequence length increases would be very clarifying.
> >
> > [1] Josh Alman, and Zhao Song. "Fast attention requires bounded entries." NeurIPS’23.
> >
> > [2] Insu Han, Rajesh Jayaram, Amin Karbasi, Vahab Mirrokni, David P. Woodruff, and Amir Zandieh. "Hyperattention: Long-context attention in near-linear time." ICLR’24.

---

> > > ### Author Response · Authors · 2024-12-04
> > > **Thank you**
> > >
> > > We sincerely thank the reviewer for acknowledging the theoretical contribution. We will try to add more experiments in our next version based on your constructive suggestions. We thank you for your valuable time for the discussion.

---

### Official Review · Reviewer_Z8SA · 2024-11-04

**Soundness:** 3
**Presentation:** 2
**Contribution:** 2
**Rating:** 5
**Confidence:** 2

**Summary:**

The authors propose a k-basis system for convolutional matrices which allows for sub-quadratic inference in attention with fewer constraints on the form of the attention as compared to previous works.

**Strengths:**

- The proposed method offers a way to reduce the prohibitive quadratic complexity of modern transformers.
- As compared to previous works, the method appears to be much less restrictive on the structure of the attention mask which allows it to be applied to modern LLM settings.

**Weaknesses:**

- The experiments section is rather light, and contains only one experiment. Given that the proposed method differs quite drastically from previous transformers, I would like to see more empirical validation and ablations on different aspects of the performance.

- It is not clear to me if the experiment which was performed on Llama3 required any training from the proposed method or not.
  - If it did not require training, I would be interested to see the performance of the proposed method which did include training or finetuning.

- The rank $k$ in k-conv is given, however after reading I am curious as to how $k$ relates to performance on long context tasks. Does $k$ need to scale commensurately with the context length?

- I am left confused by the following items in Algorithm 2:
  - What is $u$ and where is it introduced in the method?
  - What is $v$ and where is it introduced in the method?
  - What is the $s$ subscript in algorithm 2 ($\tilde{H}_s \leftarrow M_s \odot (QK^\top_s)$) supposed to denote?
  - What is happening on line 8 after achieving $\tilde{H}_s$

- The algorithmic complexity looks compelling, but what is the actual wall clock time of the experiment compared to the quadratic tranformer? Can you run an additional experiment which looks at this, even if it only utilizes random inputs? I suspect that at smaller context lengths such as 2K which is provided in the experiment, the proposed method might perform worse in terms of wall clock time, but it would be interesting to know at what length the crossover happens to where the lower complexity starts to show real gains in performance.

**Questions:**

- What relationship does the conv-basis rank ($k$) have to the context length?
- Can your method be applied on top of existing pretrained transformers without any retraining or finetuning?
- Why is the one provided experiment performed on such a small scale. Given that the method reduces the quadratic complexity of a transformer, shouldn't it be able to scale to huge context lengths without much trouble? It would be interesting to compare this to another method such as Streaming LLM [1] on a streaming perplexity task (like PG19) or other long context benchmark which scales to the order of millions of tokens.

## Overall

Overall, I think the work attempts to make an ambitious change to the attention operation in order to lower the complexity. However, I believe anyone reading this work will be left with many questions about the workings of the algorithms as well as the performance characteristics. I believe adding more experiments, ablations, and answering the questions posed here will aid in understanding the proposed method more fully.

---

> ### Author Response · Authors · 2024-11-26
>
> We express our deepest gratitude to the reviewer for the time and effort in reviewing our work. You have precisely identified the strengths of our work, namely, that our main contribution is reducing the time complexity of attention approximation without relying on restrictive assumptions, enabling successful implementation in real-world applications. Below, we address your concerns regarding our work.
>
> ### W1 & Q3 & W5: The experiments section is rather light, and contains only one experiment.
> Thank you for your question. We would like to emphasize that our work is theoretical in nature. We have provided a detailed comparison of our work with recent papers, such as [1] and [2], in Line 129–145. Our motivation stems from the fact that their assumptions are too restrictive, making their algorithms challenging to implement successfully. In contrast, our work develops an implementable, assumption-light algorithm that can be applied in real-world scenarios while maintaining a comparable running time to [1] and [2]. The experiment in our paper serves as an example to demonstrate the implementation of our theoretical contribution and to highlight its independence from the restrictive assumptions of [1]. However, it is not the primary focus of our work. Our experiment illustrates that our theoretical results can be directly implemented in a popular transformer model, Llama 3, whereas [1] lacks any experimental validation. Additionally, our work is theoretically comparable to—and, in many aspects, surpasses—the prior theoretical contributions of [1] and [2] (see Line 129–145 and our response to Reviewer vPbo for further details). We also refer the reviewer to the Global response **Empirical evaluation** section for more details.
>
> ### W2 & Q2: Training or finetuning
> Thank you for your question. For the Llama3 experiment, we do not have the training, and we directly do approximation, i.e., our methods are training-free. We don't require retraining/fine-tuning, which is one of the advantages of our method.
>
> We provide training analysis in our main analysis. However, practical training or finetuning requires resources and technical hardness. We leave them as future work.
>
> ### W3 & Q1: What relationship does the conv-basis rank (k) have to the context length?
>
> The parameter $k$ is not sensitive to the context length. Theoretically, the error bound remains the same when $k$ remains fixed while the context length increases  (see Theorem 4.4). The algorithm runtime remains almost linear time as long as $k$ is sublinearly with the length of the input token $n$. This highlights another advantage of our algorithm: it is suitable for different scenarios and offers flexibility in parameter selection. Empirically, our experiment shows a very good performance when $k = 512$.
>
> ### W4:  Confusion terms in Algorithm 2
> - Both $u$ and $v$ are local variables. They are introduced in Line 2 of Algorithm 2. They are initialized as 0 vectors. $u$ tracks the cumulative sum of conv bases, and $v$ tracks the first $T$ entries of $u$ (used for binary search, Algorithm 3).
> - $s$ in Algorithm 2 denotes the starting index for binary search, i.e., the binary search of Algorithm 3 happens between index $s$ and $t$.
> - After achieving $\tilde{H}_s$, we can get conv-basis for this index. See Lemma B.19 for more details.
>
> We thank you again for your insightful comments.

---

> > ### Comment · Reviewer_Z8SA · 2024-11-29
> > **Reviewer Response**
> >
> > Thank you for the response and answers to my questions. I understand and appreciate that this work in mainly theoretical in nature. However, after reading the work, it seems that there is one experiment comparing to a standard attention and I do not understand why more experiments could not be completed, especially on the actual runtime of the algorithm.
> >
> > I find this to be an important aspect to cover because after reading the work, there is no roadmap on where to take things from here for future research. As it stands right now to any reader, it seems like there should be no reason that this cannot be immediately applied to many experiments, as it does not require training, and can work on a pretrained transformer. Therefore, I think it is crucial to either run more experiments (latency comparisons, comparisons to other sub-quadratic attention method), or highlight the barriers that exist to doing these experiments so that future works can work on futher improvements.
> >
> > For now, I will keep my score, but I am interested to hear more from the authors on these points.

---

> > > ### Author Response · Authors · 2024-11-29
> > > **Thank you and reply to follow-up questions**
> > >
> > > We sincerely thank you for your question and your time. We would like to answer your question by providing the empirical hardness we have met.
> > >
> > > Our method has a good theoretical running time. However, in practice, our methods have slower real clock time for the following reasons.
> > > - Our algorithm needs to search the $k$-conv basis using a binary search. When we search the $i$-th basis, we need to know the $i-1$-basis location. Thus, the search steps cannot be parallelized, which takes time. Although the total complexity of our method is low, due to GPU parallelization, standard attention can be faster than us.
> > > - Our method needs to use FFT, which has benefits only when sequence/token lengths $n$ are larger than $10^4$ when $k=1$. When we choose a suitable $k$, the length needs to be larger.  Thus, our conjecture is that our methods can be faster only when $n \ge 10^6$. However, this size is beyond the capacity of our resources and most open source LLMs input sequence length.
> > > - Current LLMs have been fully optimized, e.g., FlashAttention for memory IO, which is necessary for long sequences, while our methods cannot benefit from these techniques.
> > >
> > > We hope our reply clearly answers your questions. We are willing to discuss more if the reviewer has more follow-up questions.

---

### Official Review · Reviewer_vPbo · 2024-11-04

**Soundness:** 3
**Presentation:** 3
**Contribution:** 3
**Rating:** 6
**Confidence:** 2

**Summary:**

This paper examines the possibility to compute attention using Fast Fourier Transformer when attention matrix can be well approximated by the sum of some convolution matrices. They show that when the Attention matrix can be broken down in this way, both inference and training can be done in almost linear time when $kd = n^{o(1)}$.

**Strengths:**

1. This paper investigates the timely question on how to speed up attention calculation and offers a new perspective on when attention calculation can be speed up.

2. While focusing on the theoretical perspective, this paper present empirical validation on the correctness of the presented algorithm and required $k$ to approximate attention matrix.

**Weaknesses:**

1. The algorithm 3 is deferred to appendix while being a key building block of the algorithm. In general, a more detailed explaination on how the binary search is carried out will improve the clarity of the paper.

2. Because the current work is built upon the previous work with low rank assumption, a more direct comparison (either empirically or theoretically) between the current method and previous one should be presented.

**Questions:**

1. How would the author expect $k$ to scale with context length and the size of the model?

---

> ### Author Response · Authors · 2024-11-26
>
> We express our deepest gratitude to the reviewer for taking the time and effort to review our work. It is very encouraging to see that our study is recognized for addressing a timely question, providing empirical results, and focusing on the theoretical aspects. Below, we address your concerns regarding our work.
>
> ### W1: The algorithm 3 is deferred to the appendix while being a key building block of the algorithm. In general, a more detailed explanation of how the binary search is carried out will improve the clarity of the paper.
>
> Thank you for pointing out this! We completely agree that Algorithm 3, as a key building block of our technique, is crucial. We have updated the corresponding part in the revision Line 318-323. We also made a summary below.
>
> In Algorithm 3, we use binary search to efficiently locate the convolution basis position by leveraging the non-degenerate property of the attention matrix. This allows us to find $k$-$\mathsf{conv}$-basis in our main Algorithm 1, enabling better control over the running time while bounding the error. The choice of $k$ thus balances the trade-off between accuracy and efficiency. Technically, Algorithm 3 identifies positions in the attention matrix where the $\ell_1$ norm of remaining attention values exceeds the threshold $\delta - 2 T \epsilon$. The non-degenerate property enables the binary search algorithm to find the next convolution basis position in $O(\log n)$ steps.
>
> ### W2: Because the current work is built upon the previous work with low rank assumption, a more direct comparison (either empirically or theoretically) between the current method and previous one should be presented.
>
> Thanks for pointing out this question. We would like to point out that our work does need a low-rank assumption. Note that due to Casual Mask, the attention matrix is full-rank indeed. In our paper, informally, our assumption is that the attention matrix can be decomposed into $k$-$\mathsf{conv}$-basis for some small $k$.
>
> For the theoretical comparison, we have provided in Line 129-145. For the empirical comparison, we leave it as our future work.
>
> We thank you again for your valuable suggestions.

---

> > ### Comment · Reviewer_vPbo · 2024-11-29
> >
> > I have read the response and will keep my positive rating.

---

> > > ### Author Response · Authors · 2024-11-29
> > > **Thank you**
> > >
> > > We sincerely thank your valuable suggestions and comments. We appreciate your time and positive feedback.

---

### Author Response · Authors · 2024-11-26
**Global Response**

We gratefully thank all reviewers for their valuable and constructive feedback.

We have updated a **revision** for our draft, and we have made all updates in blue. All line numbers in the rebuttal correspond to the revised version. In Line 318-323, we add a description of Algorithm 3 (binary search).

Then, we will cover some questions that reviewers commonly ask.

### Empirical evaluation
Thank you for the reviewers’ valuable suggestions to provide empirical evidence for our method. However, we would like to emphasize that our work is intended to be positioned as a theoretical study, focusing primarily on theoretical analysis. Nevertheless, to demonstrate the effectiveness of our method, we have conducted preliminary experiments in Section 6.

On the other hand, we kindly wish to emphasize that the main focus of our work is theoretical analysis. The importance of theoretical contributions is widely acknowledged in esteemed conferences such as NeurIPS, ICLR, and ICML. For example, papers [1, 2, 3, 4] accepted at ICLR are solely based on theoretical analysis without any experimental data. Likewise, studies [4, 5, 6, 7, 8] that focus on designing efficient algorithms for low-rank and attention approximations also bypass empirical results. We will leave more empirical verification in our future work.

[1] Zhan, W., Uehara, M., Kallus, N., Lee, J. D., & Sun, W. Provable Offline Preference-Based Reinforcement Learning. ICLR’24.

[2] Chen, S., Chewi, S., Li, J., Li, Y., Salim, A., & Zhang, A. R. Sampling is as easy as learning the score: theory for diffusion models with minimal data assumptions. ICLR’23.

[3] Wen, K., Ma, T., & Li, Z. How Sharpness-Aware Minimization Minimizes Sharpness?. ICLR’23.

[4]  Alman, J., & Song, Z. How to capture higher-order correlations? generalizing matrix softmax attention to kronecker computation. ICLR’24.

[5] Alman, J., & Song, Z. Fast attention requires bounded entries. NeurIPS’23.

[6] Alman, J., & Song, Z. The fine-grained complexity of gradient computation for training large language models. NeurIPS’24.

[7] Sarlos, T., Song, X., Woodruff, D., & Zhang, R. Hardness of low rank approximation of entrywise transformed matrix products. NeurIPS’24.

[8] Dexter, G., Drineas, P., Woodruff, D., & Yasuda, T. Sketching algorithms for sparse dictionary learning: PTAS and turnstile streaming. NeurIPS’24.

---

### Meta-Review · Area_Chair_mFCw · 2024-12-22

**Metareview:**

The paper proposes a theoretical framework for approximating attention using convolution matrices, offering promising computational complexity improvements. While the theoretical contributions are sound and clearly articulated, the submission suffers from significant weaknesses. Notably, empirical validation is limited to a single experiment, lacking crucial runtime or scalability benchmarks. These deficiencies undermine the paper's claims regarding practical efficiency. Additionally, key implementation details, such as comparisons with prior low-rank approaches and comprehensive ablation studies, are absent. While the theoretical findings are valuable, the lack of empirical substantiation, coupled with insufficient clarity in algorithmic details, limits its broader impact. Based on these shortcomings, I recommend rejection, encouraging the authors to enhance empirical evidence in future iterations.

**Additional Comments On Reviewer Discussion:**

Reviewers highlighted the paper's theoretical strengths but consistently emphasized the lack of empirical support, particularly regarding runtime comparisons and scaling efficiency. Although the authors acknowledged these gaps and justified their focus on theory, their response failed to address key reviewer concerns effectively. Claims of efficiency remain unsupported by runtime data or practical benchmarks. While the revisions provided some clarifications, they did not substantively resolve the concerns about applicability and validation. This limited response, combined with the paper’s theoretical focus, led to the conclusion that the submission lacks the necessary breadth for acceptance.

---

### Decision · Program_Chairs · 2025-01-22

Reject